# A Representer Theorem for Hawkes Processes via Penalized Least Squares Minimization

**Hideaki Kim & Tomoharu Iwata**
NTT, Inc.
{hideaki.kin,tomoharu.iwata}@ntt.com

## Abstract

The representer theorem is a cornerstone of kernel methods, which aim to estimate latent functions in reproducing kernel Hilbert spaces (RKHSs) in a nonparametric manner. Its significance lies in converting inherently infinite-dimensional optimization problems into finite-dimensional ones over dual coefficients, thereby enabling practical and computationally tractable algorithms. In this paper, we address the problem of estimating the latent triggering kernels–functions that encode the interaction structure between events–for linear multivariate Hawkes processes based on observed event sequences within an RKHS framework. We show that, under the principle of penalized least squares minimization, a novel form of representer theorem emerges: a family of transformed kernels can be defined via a system of simultaneous integral equations, and the optimal estimator of each triggering kernel is expressed as a linear combination of these transformed kernels evaluated at the data points. Remarkably, the dual coefficients are all analytically fixed to unity, obviating the need to solve a costly optimization problem to obtain the dual coefficients. This leads to a highly efficient estimator capable of handling large-scale data more effectively than conventional nonparametric approaches. Empirical evaluations on synthetic and real-world datasets reveal that the proposed method achieves competitive predictive accuracy while substantially improving computational efficiency compared to state-of-the-art kernel method-based estimators.

## 1 Introduction

Nonparametric estimation of latent functions remains a central topic in both theoretical and applied research, spanning domains such as signal and image processing (Liu et al., 2011; Takeda et al., 2007), system control (Liu et al., 2018), geostatistics (Chiles & Delfiner, 2012), bioinformatics (Schölkopf et al., 2004), and clinical studies (Collett, 2023). Among various nonparametric approaches, kernel methods stand out as one of the most powerful and mature frameworks. These methods enable flexible function approximation by embedding data into high-dimensional reproducing kernel Hilbert spaces (RKHSs) (Schölkopf & Smola, 2018; Shawe-Taylor & Cristianini, 2004). In classical supervised settings with i.i.d. data, the representer theorem plays a pivotal role in kernel methods. It states that the solution to a broad class of infinite-dimensional optimization problems in RKHSs admits a finite-dimensional representation: the optimal function estimator can be expressed as a linear combination of kernel functions evaluated at the training points (Schölkopf et al., 2001; Wahba, 1990). This linear form not only provides theoretical insight but also brings practical advantages in optimization and inference.

Recently, the kernel method literature has begun to address the nonparametric estimation of intensity functions in point process models. The problems are fundamentally more challenging than i.i.d. cases, primarily because the loss functions to minimize (e.g., negative log-likelihood functions) involve integrals of latent intensity functions over observation domains and violate independence assumptions, which renders classical representer theorems inapplicable. A seminal contribution by Flaxman et al. (2017) demonstrated that a representer theorem can still hold for a point process: specifically, they showed that if the square root of the intensity function lies in an RKHS, then the solution to the penalized maximum likelihood estimation problem admits a finite-dimensional

representation. Interestingly, the optimal estimator is expressed not via standard RKHS kernels but via *equivalent kernels*–RKHS kernels transformed through a Fredholm integral equation. The result has since been extended to settings with covariate-dependent intensity functions (Kim et al., 2022) and survival point processes (Kim, 2023), providing a broader foundation for kernel method-based learning in point process models.

More recently, Bonnet & Sangnier (2025) addressed a more intricate setting of multivariate Hawkes processes (Brémaud & Massoulié, 1996; Hawkes, 1971), which offer a powerful framework for modeling self- and mutually-interacting event dynamics in real-world applications such as finance (Bacry et al., 2015), neuroscience (Gerhard et al., 2017), social networks (Zhou et al., 2013), and seismology (Ogata, 1988). By leveraging the approximations of both the log-likelihood and least-squares loss functions, they obtained a representer theorem for the estimation of triggering kernels in an RKHS. To ensure the non-negativity of the intensity functions, the model employs non-linear link functions, allowing it to capture both excitatory and inhibitory interactions. Although the method demonstrates strong empirical performance, it requires solving a non-linear optimization problem over dual coefficients whose dimensionality scales with the data size, posing serious scalability issues for a large scale of datasets often seen in multivariate Hawkes processes.

In this paper, we consider a kernel method-based least squares loss formulation for estimating latent triggering kernels in linear multivariate Hawkes processes, where the identity link function is assumed. By leveraging Mercer's theorem, we establish a novel representer theorem tailored to the functional optimization problem: the obtained estimator of each triggering kernel admits a linear expansion in terms of *equivalent kernels* defined through a system of Fredholm integral equations. Notably, all dual coefficients are analytically fixed to unity, eliminating the need to solve a costly coefficient optimization problem. To the best of our knowledge, this paper is the first to establish a representer theorem for the non-approximated penalized least squares formulation of linear Hawkes processes. Furthermore, we propose an efficient algorithm to solve the integral equations using the random feature map approximation of RKHS kernels (Rahimi & Recht, 2007), where all required integrals are obtained in a closed form, in contrast to the Bonnet & Sangnier (2025) model that relies on Riemann approximation. Consequently, the proposed estimator[1] consists solely of additive matrix operations and an inversion of a matrix whose size is independent of the data size. This yields a highly lightweight and scalable estimator that remains effective even on large-scale event data, offering a practical and theoretically grounded solution for learning in multivariate Hawkes processes.

## 2 PROPOSED METHOD

### 2.1 PRELIMINARY: LINEAR HAWKES PROCESSES

We consider a multivariate linear Hawkes process (Brémaud & Massoulié, 1996; Hawkes, 1971) on a time domain $\mathbb{R}_+$, i.e., a $U$-dimensional counting process $(N_1(t), \ldots, N_U(t))$ characterized by the following conditional intensity functions:

$$\lambda_i(t) = \mu_i + \sum_{j \in \mathcal{U}} \int_0^t g_{ij}(t-s) dN_j(s), \quad t \in \mathbb{R}_+, \ i \in \mathcal{U} := [\![1, U]\!], \tag{1}$$

where $\mu_i \in \mathbb{R}_+$ denotes the baseline intensity for dimension $i$, and $g_{ij}(t-s) : \mathbb{R}_+ \to \mathbb{R}$ is the triggering kernel quantifying the change in the dimension $i$'s intensity at time $t$ caused by the event of dimension $j$ occurring at time $s$.

Let $\{(t_n, u_n) \in \mathbb{R}_+ \times \mathcal{U}\}_{n=1}^{N(T)}$ denote a sequence of $N(T) = \sum_{i \in \mathcal{U}} N_i(T)$ observed events over an interval $[0, T]$, where each pair $(t_n, u_n)$ indicates that the $n$-th event occurred at time $t_n$ on dimension $u_n$. In the literature on point processes, two common approaches have been used to estimate the intensity functions: one based on the negative log-likelihood function (Daley & Vere-Jones, 2006), and the other on the least squares contrast (Hansen et al., 2015), defined respectively as

$$L_{\text{LL}} = \sum_{i \in \mathcal{U}} \left[ \int_0^T \lambda_i(t) dt - \sum_{n \in \mathcal{N}_i} \log\big(\lambda_i(t_n)\big) \right], \quad L_{\text{LS}} = \sum_{i \in \mathcal{U}} \left[ \int_0^T \lambda_i(t)^2 dt - 2 \sum_{n \in \mathcal{N}_i} \lambda_i(t_n) \right], \tag{2}$$

---

[1]Codes are available at: `https://github.com/HidKim/K2Hawkes`

where $\mathcal{N}_i = \{n : u_n = i\}_{n=1}^{N(T)}$ denotes a subset of event indices associated with dimension $i$. Notably, the least squares contrast, $L_{\text{LS}}$, arises from the principle of empirical risk minimization (van de Geer, 2000), and has recently attracted attention due to its favorable computational properties in Hawkes process modeling (Bacry et al., 2020; Cai et al., 2024).

## 2.2 A Representer Theorem for Linear Hawkes Processes

Let $k : \mathcal{T} \times \mathcal{T} \to \mathbb{R}$ be a positive semi-definite kernel on a one-dimensional compact space $\mathcal{T} \subset \mathbb{R}$. Then there exists a unique reproducing kernel Hilbert space (RKHS) $\mathcal{H}_k$ (Schölkopf & Smola, 2018; Shawe-Taylor & Cristianini, 2004) associated with RKHS kernel $k(\cdot, \cdot)$.

Given an observed sequence of events $\{(t_n, u_n)\}_{n=1}^{N(T)}$ over an interval $[0, T]$, we consider the following regularized optimization problem of triggering kernels, $g = \{g_{ij}(\cdot)\}_{(i,j) \in \mathcal{U}^2}$, and baseline intensities, $\mu = \{\mu_i\}_{i \in \mathcal{U}}$, in the linear Hawkes process (1):

$$\hat{g}, \hat{\mu} = \operatorname*{arg\,min}_{g \in \mathcal{H}_k^{U^2}, \ \mu \in \mathbb{R}^U} \left[ L(g, \mu) + \frac{1}{\gamma} \sum_{(i,j) \in \mathcal{U}^2} \|g_{ij}\|_{\mathcal{H}_k}^2 \right], \tag{3}$$

where $L$ represents the loss functional, $\| \cdot \|_{\mathcal{H}_k}^2$ represents the squared Hilbert space norm, and $\gamma \in \mathbb{R}_+$ represents the regularization hyperparameter. In this paper, we adopt the least squares contrast for point processes, denoted by $L_{\text{LS}}(g, \mu)$, as a loss functional, which takes a quadratic form in terms of the triggering kernels and baseline intensities,

$$\begin{aligned} L_{\text{LS}}(g, \mu) = \sum_{i \in \mathcal{U}} \bigg[ &\int_0^T \Big( \mu_i + \sum_{n \in \mathcal{N}} g_{iu_n}(t - t_n) \mathbf{1}_{0 < t - t_n \le A} \Big)^2 dt \\ &- 2 \sum_{n' \in \mathcal{N}_i} \Big( \mu_i + \sum_{n \in \mathcal{N}} g_{iu_n}(t_{n'} - t_n) \mathbf{1}_{0 < t_{n'} - t_n \le A} \Big) \bigg], \end{aligned} \tag{4}$$

where $\mathbf{1}_{(\cdot)}$ denotes the indicator, and $\mathcal{N} = \{n\}_{n=1}^{N(T)}$ represents the whole indices of events observed. Here, a finite support window $A \in \mathbb{R}_+$ for the triggering kernels is introduced, as is commonly done in Hawkes process modeling to reduce computational cost (Bonnet et al., 2023; Halpin, 2013). Theorem 1 establishes a novel representer theorem for the functional optimization problem defined in Equations (3-4). Notably, all dual coefficients are analytically fixed to unity, eliminating the need to optimize them. We present a proof sketch based on Mercer's theorem (Mercer, 1909), with the complete proof deferred to Appendix B. For additional intuition, we also provide an alternative derivation via the path integral representation (Kim, 2021) in Appendix C.

**Theorem 1.** *Given the estimation of the baseline intensity $\{\hat{\mu}_i\}_{i \in \mathcal{U}}$, the solutions of the functional optimization problem (3-4), denoted as $\{\hat{g}_{ij}(\cdot)\}_{(i,j) \in \mathcal{U}^2}$, involve the representer theorem under a set of equivalent kernels[2], $\{h_j(\cdot, \cdot)\}_{j \in \mathcal{U}}$, and their dual coefficients are equal to unity:*

$$\hat{g}_{ij}(s) = \sum_{n \in \mathcal{N}_i} \alpha_n^{ij} h_j(s, t_n) - \hat{\mu}_i \int_0^T h_j(s, t) dt, \quad \alpha_n^{ij} = 1, \ s \in \mathcal{T}, \ (i, j) \in \mathcal{U}^2, \tag{5}$$

*where $\{\alpha_n^{ij} = 1\}$ denote the dual coefficients, and the equivalent kernels $\{h_j(\cdot, \cdot)\}_{j \in \mathcal{U}}$ are defined through a system of Fredholm integral equations,*

$$\frac{1}{\gamma} h_j(s, s') + \sum_{l \in \mathcal{U}} \int_0^T V_{jl}(s, t) h_l(t, s') dt = \sum_{n \in \mathcal{N}_j} k(s, s' - t_n) \mathbf{1}_{0 < s' - t_n \le A},$$

$$V_{jl}(s, t) = \sum_{n \in \mathcal{N}_j} \sum_{n' \in \mathcal{N}_l} k(s, t + t_{n'} - t_n) \mathbf{1}_{\max(t_n, t_{n'}) < t + t_{n'} \le \min(T, A + t_n, A + t_{n'})}. \tag{6}$$

*Proof sketch.* Through Mercer's theorem, the RKHS kernel $k(\cdot, \cdot)$ can be expressed through its Mercer's expansion:

$$k(t, s) = \sum_{m=1}^{\infty} e_m(t) e_m(s), \quad \int_{\mathcal{T}} e_m(t) e_{m'}(t) dt = \eta_m \delta_{mm'}, \tag{7}$$

---

[2]Following Flaxman et al. (2017), we call the transformed kernel functions where a representer theorem holds the *equivalent kernels*.

where $\{e_m(\cdot)\}_{m=1}^{\infty}$ and $\{\eta_m\}_{m=1}^{\infty}$ denote the eigenfunctions and the eigenvalues, respectively, of the integral operator $\int_{\mathcal{T}} \cdot \, k(t,s)ds$. Accordingly, the triggering kernels in the RKHS, $\{g_{ij}(\cdot) \in \mathcal{H}_k\}_{(i,j)\in\mathcal{U}^2}$, and their squared RKHS norms, $\|g_{ij}\|_{\mathcal{H}_k}^2$, admit the representation

$$g_{ij}(s) = \sum_{m=1}^{\infty} b_{ij}^m \, e_m(s), \qquad \|g_{ij}\|_{\mathcal{H}_k}^2 = \sum_{m=1}^{\infty} (b_{ij}^m)^2, \qquad (i,j) \in \mathcal{U}^2, \tag{8}$$

where $b = \{b_{ij}^m \in \mathbb{R}\}$ is the expansion coefficient. Using this representation, the optimization problem (3-4) can be reformulated as follows:

$$\hat{b}, \ \hat{\mu} = \arg\min_{b, \ \mu} \left[ L_{\mathrm{LS}}(b,\mu) + \frac{1}{\gamma} \sum_{(i,j)\in\mathcal{U}^2} \sum_{m=1}^{\infty} (b_{ij}^m)^2 \right], \tag{9}$$

where

$$
\begin{aligned}
L_{\mathrm{LS}}(b,\mu) = \sum_{i\in\mathcal{U}} \Bigg[ & \int_0^T \Big( \mu_i + \sum_{n\in\mathcal{N}} \sum_{m=1}^{\infty} b_{iu_n}^m e_m(t-t_n) \mathbf{1}_{0<t-t_n\leq A} \Big)^2 dt \\
& - 2 \sum_{n'\in\mathcal{N}_i} \Big( \mu_i + \sum_{n\in\mathcal{N}} \sum_{m=1}^{\infty} b_{iu_n}^m e_m(t_{n'}-t_n) \mathbf{1}_{0<t_{n'}-t_n\leq A} \Big) \Bigg].
\end{aligned}
\tag{10}
$$

Given the estimate of the baseline intensity $\hat{\mu}$, the optimal coefficient vector $\hat{b}$ must satisfy the equation obtained by setting the gradient of the objective with respect to $b$ equal to zero:

$$\frac{\partial}{\partial b_{ij}^m} \left[ L_{\mathrm{LS}}(b,\hat{\mu}) + \frac{1}{\gamma} \sum_{(i,j)\in\mathcal{U}^2} \sum_{m'=1}^{\infty} (b_{ij}^{m'})^2 \right]\Bigg|_{b=\hat{b}} = 0, \quad (i,j) \in \mathcal{U}^2, \ m \in \{1,2,\dots\}. \tag{11}$$

Using the Mercer's expansion (7) and the kernel trick, $k(t,s) = \sum_m e_m(t)e_m(s)$, Equation (11) leads to Equations (5-6). $\blacksquare$

Note that for any positive semi-definite kernel $k(\cdot,\cdot)$, the corresponding set of equivalent kernels $\{h_j(\cdot,\cdot)\}_{j\in\mathcal{U}}$ defined in Equation (6) exists and is uniquely defined. See Appendix G for the proof.

While Bonnet & Sangnier (2025) has also explored representer theorems under the least squares contrast for linear Hawkes processes, their formulation relies on a discretized approximation (see Proposition 1 in (Bonnet & Sangnier, 2025)), which introduces additional optimization over dual coefficients and obscures the elegant mathematical properties of the least squares contrast. To the best of our knowledge, this work is the first to establish a representer theorem for the non-discretized penalized least squares formulation of linear Hawkes processes.

In Theorem 1, the optimal estimators of baseline intensities, $\{\hat{\mu}_i\}_{i\in\mathcal{U}}$ are treated as given constants. Proposition 2 demonstrates that by substituting Equation (5) into Equation (3), $\{\hat{\mu}_i\}_{i\in\mathcal{U}}$ can be obtained in closed form in terms of the equivalent kernels. The proof is provided in Appendix D.

**Proposition 2.** *The solutions, $\{\hat{\mu}_i\}_{i\in\mathcal{U}}$, of the functional optimization problem (3-4) have closed forms in terms of the equivalent kernels defined by Equation (6) as follows:*

$$\hat{\mu}_i = \frac{|\mathcal{N}_i| - \sum_{n\in\mathcal{N}} \sum_{n'\in\mathcal{N}_i} \int_0^T h_{u_n}(t-t_n, t_{n'}) \mathbf{1}_{0<t-t_n\leq A} dt}{T - \sum_{n\in\mathcal{N}} \int_0^T \int_0^T h_{u_n}(t-t_n, s) \mathbf{1}_{0<t-t_n\leq A} dtds}, \qquad i \in \mathcal{U}, \tag{12}$$

*where $|\mathcal{N}_i|$ denotes the number of observed events associated with dimension $i$.*

### 2.3 CONSTRUCTION OF EQUIVALENT KERNELS

In Section 2.2, we showed that the optimal estimators of $g$ and $\mu$ can be expressed in closed form using the equivalent kernels $\{h_j(\cdot,\cdot)\}_{j\in\mathcal{U}}$. However, obtaining $\{h_j(\cdot,\cdot)\}_{j\in\mathcal{U}}$ in practice requires solving the coupled integral equations (6), which is generally a non-trivial task. In Proposition 3, we propose a solution based on the degenerate kernel approximation methods (Atkinson, 2010; Polyanin & Manzhirov, 1998). The proof is provided in Appendix E.

**Proposition 3.** *Let an RKHS kernel $k(\cdot, \cdot)$ have a degenerate form with $M$ feature maps $\{\phi_m(s)\}$,*

$$k(s, s') = \sum_{m=1}^{M} \phi_m(s)\phi_m(s') = \phi(s)^\top \phi(s'), \tag{13}$$

*where $\phi(s) = (\phi_1(s), \ldots, \phi_M(s))^\top$. Then the solution of the simultaneous Fredholm integral equations (6) can be obtained in closed form as follows:*

$$h_j(s, s') = \phi(s)^\top \left[ \left( \frac{1}{\gamma} \boldsymbol{I}_{MU} + \boldsymbol{\Xi} \right)^{-1} \tilde{\phi}(s') \right]_{1+(j-1)M:jM}, \quad j \in \mathcal{U}, \tag{14}$$

*where $[\cdot]_{a:b}$ denotes the slice of matrix between the $a$-th row and the $b$-th one, $\boldsymbol{I}_{MU} \in \mathbb{R}^{MU \times MU}$ denotes the identity matrix, $\boldsymbol{\Xi} = [\boldsymbol{\Xi}_{ij}] \in \mathbb{R}^{MU \times MU}$ is defined as a symmetric block matrix whose $(i, j)$-th block is given by an $M$-by-$M$ submatrix,*

$$\boldsymbol{\Xi}_{ij} = \sum_{n \in \mathcal{N}_i} \sum_{n' \in \mathcal{N}_j} \mathbf{1}_{\max(t_n, t_{n'}) < \min(T, A+t_n, A+t_{n'})} \int_{\max(t_n, t_{n'})}^{\min(T, A+t_n, A+t_{n'})} \phi(t - t_n)\phi(t - t_{n'})^\top dt, \tag{15}$$

*and $\tilde{\phi}(s) : \mathcal{T} \to \mathbb{R}^{MU}$ denotes a concatenated vector function,*

$$\tilde{\phi}(s) = \left[ \tilde{\phi}_1(s) \mid \tilde{\phi}_2(s) \mid \ldots \mid \tilde{\phi}_U(s) \right], \quad \tilde{\phi}_i(s) = \sum_{n \in \mathcal{N}_i} \phi(s - t_n)\mathbf{1}_{0 < s - t_n \le A}. \tag{16}$$

Proposition 4 shows that substituting the equivalent kernels (14) into Equations (5) and (12), we can obtain the optimal estimators in terms of the feature maps. The proof is provided in Appendix F.

**Proposition 4.** *For a degenerate form of RKHS kernel in (13), the optimal estimators, $\hat{g}$ and $\hat{\mu}$, are obtained in closed form in terms of the feature maps:*

$$\hat{g}_{ij}(s) = \phi(s)^\top \left[ \left( \frac{1}{\gamma} \boldsymbol{I}_{MU} + \boldsymbol{\Xi} \right)^{-1} \left( \sum_{n \in \mathcal{N}_i} \tilde{\phi}(t_n) - \hat{\mu}_i \int_0^T \tilde{\phi}(t)dt \right) \right]_{1+(j-1)M:jM},$$

$$\hat{\mu}_i = \frac{|\mathcal{N}_i| - \left( \int_0^T \tilde{\phi}(t)dt \right)^\top \left( \frac{1}{\gamma} \boldsymbol{I}_{MU} + \boldsymbol{\Xi} \right)^{-1} \left( \sum_{n \in \mathcal{N}_i} \tilde{\phi}(t_n) \right)}{T - \left( \int_0^T \tilde{\phi}(t)dt \right)^\top \left( \frac{1}{\gamma} \boldsymbol{I}_{MU} + \boldsymbol{\Xi} \right)^{-1} \left( \int_0^T \tilde{\phi}(t)dt \right)}. \tag{17}$$

In this paper, we assume that RKHS kernels are shift-invariant, i.e., $k(s, s') = k(|s - s'|)$, which includes popular kernels such as Gaussian, Matérn, and Laplace kernels. We employ the random Fourier feature method (Rahimi & Recht, 2007), approximating the shift-invariant RKHS kernel as a sum of Fourier features sampled from the Fourier transform of the kernel, denoted by $\tilde{k}(\omega)$, as

$$\phi_m(s) = \sqrt{\frac{2}{M}} \cos(\omega_m s + \theta_m), \quad \omega_m = \begin{cases} \sim \tilde{k}(\omega) & m \le \frac{M}{2} \\ \omega_{m-\frac{M}{2}} & m > \frac{M}{2} \end{cases}, \quad \theta_m = \begin{cases} 0 & m \le \frac{M}{2} \\ -\frac{\pi}{2} & m > \frac{M}{2} \end{cases}. \tag{18}$$

To enhance the approximation accuracy of the random Fourier features, we employed the quasi-Monte Carlo feature maps (Yang et al., 2014), and used $M = 100$ in Section 4. Then the integral operations appeared in (15) and (17) can be performed analytically as follows:

$$\int_0^T \tilde{\phi}_i(s)ds = \sqrt{\frac{2}{M}} \frac{1}{\boldsymbol{\omega}} \circ \left[ \sin(\boldsymbol{\omega} \cdot \min(T, A - t_n) + \boldsymbol{\theta}) - \sin(\boldsymbol{\theta}) \right] \in \mathbb{R}^M,$$

$$\int_a^b \phi(t - t_n)\phi(t - t_{n'})^\top dt = \frac{b - a}{M} \left[ \zeta(\boldsymbol{\omega}, \boldsymbol{\omega}^\top, \boldsymbol{\theta}, \boldsymbol{\theta}^\top) + \zeta(\boldsymbol{\omega}, -\boldsymbol{\omega}^\top, \boldsymbol{\theta}, -\boldsymbol{\theta}^\top) \right] \in \mathbb{R}^{M \times M}, \tag{19}$$

where $\circ$ denotes the Hadamard product, $\boldsymbol{\omega} = (\omega_1, \ldots, \omega_M)^\top$, $\boldsymbol{\theta} = (\theta_1, \ldots, \theta_M)^\top$, and

$$\zeta(\omega, \omega', \theta, \theta') = \cos\left[ (b + a)(\omega + \omega')/2 + \theta + \theta' - \omega t_n - \omega' t_{n'} \right] \text{sinc}\left[ (b - a)(\omega + \omega')/2 \right]. \tag{20}$$

Here, $\text{sinc}(x) = \sin(x)/x$ is the unnormalized sinc function. As a result, the optimal estimators are obtained in closed form without requiring any discretization approximation of the integral operators. It is worth noting that the number of feature maps, $M$, required to approximate a one-dimensional RKHS kernel remains modest regardless of the data size, whereas the number of discretization nodes needed for accurate integral evaluations grows with the data size. See Section 3 for details.

### 2.4 COMPLEXITY ANALYSIS

The computational complexity of obtaining our estimators in Equation (17) is $\mathcal{O}(\underline{N}^2 M^2 U^2 + M^3 U^3)$, where $\underline{N} = \max(|\mathcal{N}_1|, \ldots, |\mathcal{N}_U|)$: the first term arises from the computation of $\boldsymbol{\Xi}$, and the second from the inversion of $\left(\gamma^{-1}\boldsymbol{I}_{MU} + \boldsymbol{\Xi}\right)$. Its memory complexity is $\mathcal{O}(M^2 U^2)$, which stems of $\boldsymbol{\Xi}$. In contrast, the prior kernel method-based estimator (Bonnet & Sangnier, 2025) requires the computation of $\mathcal{O}(\underline{N}^4 U^2 P)$, where $P$ denotes the number of iterations needed for convergence in an iterative optimization algorithm. Its memory complexity is $\mathcal{O}(\underline{N}^2 U^2)$. Therefore, our approach achieves significantly better scalability with respect to the data size compared to the previous method, making it well-suited for large-scale data scenarios. Moreover, our method requires only a single matrix inversion and avoids the need to carefully tune convergence criteria and learning rates, offering a more stable and practical solution, which is in contrast to the prior kernel method-based method that relies on iterative optimization.

## 3 RELATED WORK

Hawkes processes (Hawkes, 1971), particularly in the multivariate setting, have been extensively studied due to their expressive power in modeling self- and mutually-exciting temporal dynamics on networks. One of the simplest approaches to learning the triggering kernels in Hawkes processes is parametric modeling, where exponential kernels are particularly popular owing to their ability to encode interaction strength and temporal decay compactly. In the case of linear Hawkes processes (1), maximum likelihood estimation has been the gold standard (Bacry et al., 2015; Ozaki, 1979; Zhou et al., 2013). However, several alternative estimation strategies have been proposed, including least squares-based approaches that exploit analytic tractability (Bacry et al., 2020), spectral methods (Adamopoulos, 1976), and moment-matching methods (Da Fonseca & Zaatour, 2014).

Most of the above models assume mutual excitation, i.e., non-negative triggering kernels. Recently, however, Bonnet et al. (2023) introduced a flexible non-linear Hawkes model (see Equation (21)) with exponential triggering kernels, which enables us to estimate both excitatory and inhibitory interactions efficiently within exponential forms. While the complexity of model fitting scales linearly with the number of events, the model requires processing event times successively to evaluate the likelihood function and is therefore difficult to parallelize across multiple cores, limiting scalability.

In the nonparametric regime, a wide variety of methods have been explored for linear Hawkes processes, which include piece-wise constant (Reynaud-Bouret et al., 2014) and Gaussian mixture (Xu et al., 2016) representations of triggering kernels, and the estimation method via the solution of Wiener–Hopf equations (Bacry & Muzy, 2016). For non-linear Hawkes processes, nonparametric formulations such as those using Bernstein-type polynomials (Lemonnier & Vayatis, 2014) and B-spline expansions (Cai et al., 2024) have been proposed. Additionally, many neural network-based models have been developed to learn event dynamics directly from data, ranging from RNN-based approaches (Mei & Eisner, 2017) to Transformers (Zuo et al., 2020). For a more comprehensive survey, we refer readers to (Bonnet & Sangnier, 2025; Bonnet et al., 2023).

While prior works using reproducing kernel Hilbert spaces (RKHSs) remain relatively underexplored in the context of Hawkes processes, two notable exceptions exist. Yang et al. (2017) propose an online estimation method in RKHS under a regret minimization framework, which fundamentally differs from the batch learning setting considered in this paper. Bonnet & Sangnier (2025), on the other hand, considered a non-linear multivariate Hawkes process,

$$\lambda_i(t) = \varphi\left(\mu_i + \sum_{j \in \mathcal{U}} \int_0^t g_{ij}(t-s)dN_j(s)\right), \quad t \in \mathbb{R}_+, \ i \in \mathcal{U} := [\![1, U]\!], \quad (21)$$

where $\varphi(x) = \log(1 + e^{wx})/w$ is a non-negative soft-plus function ($w = 100$), and demonstrated a representer theorem for the problem by adopting an approximate likelihood function (Lemonnier & Vayatis, 2014) or an upper bound on the least squares loss (Lemonnier & Vayatis, 2014) as the objective. Specifically, they showed that in both cases, the optimal estimator of each triggering kernel admits a linear representation in terms of RKHS kernels as follows:

$$\hat{g}_{ij}(\cdot) = \alpha_0^{ij} \sum_{n \in \mathcal{N}_j} \int_0^T k(\cdot, t-t_n)\mathbf{1}_{0 < t-t_n \leq A}dt + \sum_{n \in \mathcal{N}_i} \alpha_1^{nj} \sum_{n' \in \mathcal{N}_j} k(\cdot, t_n - t_{n'})\mathbf{1}_{0 < t_n - t_{n'} \leq A}, \quad (22)$$

where $\{\alpha_0^{ij}, \alpha_1^{nj}\}$ denote dual coefficients of dimension $U(N(T) + U)$. Rather than solving the associated dual optimization problem directly, they adopted the linear representation (22) as a semi-parametric model and estimated the dual coefficients by maximizing the objective in (3), where the integrals in the loss functionals (2) were evaluated via discretization approximation. For a detailed formulation of the resulting objective function, see Section 3.3 of (Bonnet & Sangnier, 2025).

While the approach (21-22) exhibits strong empirical performance, it involves solving a non-linear optimization problem over dual coefficients, which demands the computation of $\mathcal{O}(\underline{N}^4 U^2)$ for each evaluation of the objective function, resulting in significant scalability challenges for large-scale datasets, as is often the case in multivariate Hawkes processes. What makes the situation worse is that the intensity functions in Hawkes processes are usually discontinuous at the observed event times, and to accurately evaluate the integrals of the intensity functions in the loss functionals (2), a dense set of discretization nodes is required as the number of events increases. As a result, the computational cost can grow significantly as the dataset becomes larger.

Here, we have primarily focused on kernel method-based Hawkes processes. For a discussion of their relationship to another major approach, i.e., deep neural network models, we refer the reader to Appendix I.

## 4 EXPERIMENTS

We evaluated the validity of our proposed method (`Ours`) by comparing it with prior parametric and non-parametric approaches. In accordance with Bonnet & Sangnier (2025), we adopted the following four approaches as baselines: `Exp` is the state-of-the-art parametric approach based on a non-linear Hawkes process with exponential triggering kernels (Bonnet et al., 2023); `Gau` is a non-parametric approach based on a linear Hawkes process (Xu et al., 2016), where the triggering kernels are represented as Gaussian mixtures; `Ber` is a non-parametric approach based on a non-linear Hawkes process (Lemonnier & Vayatis, 2014), where the triggering kernels are represented as Bernstein-type polynomials; `Bonnet` is the state-of-the-art kernel method-based approach (Bonnet & Sangnier, 2025), which assumes the triggering kernels lie in an RKHS. For `Bonnet`, the node size of discretization approximation was set at $\max(1000, 2|\mathcal{N}_1|, \ldots, 2|\mathcal{N}_U|)$ following (Bonnet & Sangnier, 2025). For `Ours` and `Bonnet`, a Gaussian RKHS kernel was employed: $k(s, s') = e^{-(\beta|s - s'|)^2}$, where $\beta$ is the inverse scale hyperparameter. For `Gau` and `Ber`, the number of basis functions was set at 50. For the models except `Exp`, the support window $A$ was set at 5.

The four baselines were implemented using the Python code in Bonnet & Sangnier (2025) (MIT License), and our model using TensorFlow-2.10 (Abadi et al., 2015). All the experiments were executed on a MacBook Pro equipped with a 12-core CPU (Apple M2 Max), with the GPU disabled.

### 4.1 SYNTHETIC DATA

Except for `Exp`, the models have hyperparameters to optimize. `Gau` and `Ber` have the regularization hyperparameter $\gamma$ for a quadratic penalty on the coefficients of mixture models, and `Ours` and `Bonnet` have the inverse scale hyperparameter $\beta$ in addition to $\gamma$. We optimized the hyperparameters on the grids of $\gamma \in \{0.1, 0.5, 1.0\}$ and $\beta \in \{0.5, 1.0, 1.5\}$, based on the negative log-likelihood (`Gau`, `Ber`, `Bonnet`) and the least squares loss (`Ours`) minimization. Specifically, for a sequence of events observed in an interval $[0, T]$, each model was fitted with the events in $[0, 0.8T]$, evaluated the negative log-likelihood/least squares loss for the rest of the data in $[0.8T, T]$, and the hyperparameters to minimize the criteria were chosen.

Predictive performance was assessed using the integrated squared error ($\Delta^2$) defined as follows:

$$\Delta^2 = \sum_{i \in \mathcal{U}} \sum_{j \in \mathcal{U}} \int_0^A |g_{ij}(s) - \hat{g}_{ij}(s)|^2 ds, \tag{23}$$

where $A = 5$, and $g_{ij}(s)$ and $\hat{g}_{ij}(s)$ denote the true and estimated triggering kernels, respectively. Efficiency was evaluated based on the CPU time, denoted by $cpu$, required to execute the model fitting given the optimized hyperparameters.

Table 1: Results of `Exp` (Bonnet et al., 2023), `Gau` (Xu et al., 2016), `Ber` (Lemonnier & Vayatis, 2014), `Bonnet` (Bonnet & Sangnier, 2025), and `Ours` on mutually-exciting scenario dataset across 10 trials with standard errors in brackets. $\tilde{N}$ is the average data size per trial. $cpu$ is the CPU time in seconds. The performances not significantly ($p \geq 0.01$) different from the best one under the Mann-Whitney U test (Holm, 1979) are shown in bold.

| | | Exp | | Gau | | Ber | | Bonnet | | Ours | |
|---|---|---|---|---|---|---|---|---|---|---|---|
| $T$ | $\tilde{N}$ | $\Delta^2$ | $cpu$ | $\Delta^2$ | $cpu$ | $\Delta^2$ | $cpu$ | $\Delta^2$ | $cpu$ | $\Delta^2$ | $cpu$ |
| 2000 | 1318 | 0.29 | 46.1 | **0.20** | **3.42** | 0.51 | **4.06** | **0.21** | 135 | 0.38 | **3.16** |
| | | (0.01) | (23.3) | (0.02) | (1.33) | (0.16) | (2.03) | (0.05) | (144) | (0.15) | (2.42) |
| 3000 | 2055 | 0.29 | 74.7 | **0.19** | **4.94** | 0.69 | **6.61** | **0.19** | 272 | 0.27 | **4.76** |
| | | (0.00) | (41.3) | (0.02) | (3.00) | (0.50) | (4.71) | (0.04) | (298) | (0.06) | (3.97) |
| 5000 | 4081 | 0.28 | 134 | **0.18** | **12.2** | 0.30 | **19.0** | **0.15** | 1358 | **0.20** | **10.7** |
| | | (0.00) | (64.9) | (0.01) | (8.91) | (0.06) | (15.4) | (0.02) | (1270) | (0.06) | (7.71) |
| 7000 | 5380 | 0.28 | 180.4 | **0.18** | **17.3** | 0.27 | 29.0 | **0.14** | 2070 | **0.16** | **13.1** |
| | | (0.00) | (56.6) | (0.02) | (8.00) | (0.04) | (16.1) | (0.04) | (1210) | (0.04) | (5.55) |

### 4.1.1 MUTUALLY-EXCITING SCENARIO

We consider synthetic data generated from a 3-variate linear Hawkes process (1) with baseline intensities, $\mu_i = 0.01$ for $i \in \mathcal{U}$, and mutually-exciting triggering kernels ($g_{ij} > 0$) defined as follows:

$$
\begin{array}{lll}
g_{11}(s) = 0.5e^{-s} & g_{12}(s) = 0.5e^{-10(s-1)^2} & g_{13}(s) = 0.5e^{-20(s-3)^2} \\
g_{21}(s) = 2^{-5s-1} & g_{22}(s) = 0.3e^{-0.5s} & g_{23}(s) = 0.5e^{-20(s-2)^2} \quad , \\
g_{31}(s) = 0.2e^{-3(s-2)^2} & g_{32}(s) = 0.25(1+\cos(\pi s))e^{-s} & g_{33}(s) = 0.5e^{-s}
\end{array}
\tag{24}
$$

of which setting is a modification of the one that appeared in Bonnet & Sangnier (2025). We simulated 10 trial sequences of events over the interval $[0, T]$ and performed the estimation of triggering kernels 10 times using the compared methods. To clarify the model efficiency regarding data size, we set the horizon at $T \in [2000, 3000, 5000, 7000]$.

Table 1 displays the predictive error and computational efficiency on the mutually-exciting scenario dataset. Some estimation results are displayed in Appendix A. The results demonstrate that, given a sufficiently large amount of data, our proposal achieved comparable predictive accuracy to `Bonnet`, the SOTA kernel method-based approach, while requiring several orders of magnitude less computation time for model fitting. In small data regimes, `Bonnet` tended to achieve higher accuracy than `Ours`, which may be attributed to the theoretical advantage of negative log-likelihood over least squares loss in reducing estimation biases (Bacry et al., 2016). `Gau` consistently achieved high predictive accuracy with small computation time, because it is the only baseline specifically designed for mutually-exciting interactions, aligned with the underlying process. `Exp` performed well only for the exponential triggering kernels $\{g_{ii}(\cdot)\}_{i \in \mathcal{U}}$ (see Figure 1).

We additionally conducted experiments to investigate the relationship between the estimation accuracy of the latent triggering kernels (measured by $\Delta^2$) and the predictive performance of the point process model (measured by the negative log-likelihood). The results are provided in Appendix H.4.

### 4.1.2 REFRACTORY SCENARIO

Refractory phenomena arise in point processes exhibiting short-term self-inhibition, where the occurrence of an event temporarily suppresses the likelihood of subsequent events. Such behavior is observed in neuronal spike trains (Berry & Meister, 1997) and in sequences of mainshock events (Rotondi & Varini, 2019). Here, we consider synthetic data generated from a 3-variate non-linear Hawkes process (21) with short-term self-inhibition adopted in (Bonnet & Sangnier, 2025),

$$
\begin{aligned}
g_{11}(s) &= (8s^2 - 1)\mathbf{1}_{s \leq 0.5} + e^{-2.5(s-0.5)}\mathbf{1}_{s > 0.5}, \\
g_{22}(s) &= g_{33}(s) = (8s^2 - 1)\mathbf{1}_{s \leq 0.5} + e^{-(s-0.5)}\mathbf{1}_{s > 0.5},
\end{aligned}
\tag{25}
$$

and various non-inhibitory inter-interactions,

$$
\begin{array}{lll}
g_{12}(s) = 0.6e^{-10(s-1)^2} & g_{13}(s) = 0.8e^{-20(s-3)^2} & g_{21}(s) = 0.6 \cdot 2^{-5s} \\
g_{23}(s) = 0.8e^{-20(s-2)^2} & g_{31}(s) = 0 & g_{32}(s) = 0
\end{array}
\quad .
\tag{26}
$$

Table 2: Results on refractory scenario dataset across 10 trials. Notations follow Table 1.

| | | Exp | | Gau | | Ber | | Bonnet | | Ours | |
|---|---|---|---|---|---|---|---|---|---|---|---|
| $T$ | $\tilde{N}$ | $\Delta^2$ | $cpu$ | $\Delta^2$ | $cpu$ | $\Delta^2$ | $cpu$ | $\Delta^2$ | $cpu$ | $\Delta^2$ | $cpu$ |
| 2000 | 2050 | 2.19 | 124 | 1.51 | **7.03** | 1.23 | **10.8** | **0.63** | 413 | 0.95 | **5.04** |
| | | (0.03) | (61.3) | (0.02) | (3.58) | (0.31) | (6.19) | (0.19) | (291) | (0.24) | (3.92) |
| 3000 | 2956 | 2.19 | 183 | 1.50 | **10.8** | 0.96 | 18.5 | **0.56** | 927 | **0.85** | **7.66** |
| | | (0.02) | (47.9) | (0.02) | (4.11) | (0.14) | (7.58) | (0.22) | (460) | (0.19) | (4.74) |
| 5000 | 5222 | 2.20 | 355 | 1.47 | **24.7** | 0.82 | 44.0 | **0.44** | 3197 | **0.59** | **14.9** |
| | | (0.02) | (74.0) | (0.01) | (8.41) | (0.16) | (15.8) | (0.18) | (1323) | (0.13) | (5.48) |
| 7000 | 6887 | 2.20 | 503 | 1.47 | 37.3 | 0.71 | 74.7 | **0.41** | 5884 | **0.50** | **16.1** |
| | | (0.02) | (108) | (0.01) | (14.5) | (0.09) | (29.0) | (0.19) | (2972) | (0.07) | (5.26) |

Table 3: Performance of `Ours` regarding the support window $A$ on mutually-exciting scenario dataset ($T = 5000$) across 10 trials with standard errors in brackets.

| $A = 1$ | | $A = 2$ | | $A = 5$ | | $A = 10$ | | $A = 20$ | |
|---|---|---|---|---|---|---|---|---|---|
| $\Delta^2$ | $cpu$ | $\Delta^2$ | $cpu$ | $\Delta^2$ | $cpu$ | $\Delta^2$ | $cpu$ | $\Delta^2$ | $cpu$ |
| 0.51 | 1.75 | 0.24 | 3.89 | 0.20 | 10.7 | 0.29 | 22.5 | 0.33 | 39.1 |
| (0.04) | (1.24) | (0.03) | (2.74) | (0.06) | (7.71) | (0.11) | (16.9) | (0.07) | 32.4 |

The remaining experimental conditions follow those of the mutually-exciting scenario. We adopted the soft-plus function as the link function, which is consistent with the assumption in `Bonnet`.

Table 2 presents the predictive error and computational efficiency on the refractory scenario dataset. Some estimation results are displayed in Figure 2. As shown, `Ours` was outperformed by `Bonnet` in terms of accuracy on small datasets, but the gap became less significant as the dataset size increased. In contrast to the mutually-exciting scenario, `Gau` performed poorly here due to its inability to model inhibitory interaction. While `Ber` succeeded in reconstructing the inhibitory interactions, it still fell short of the kernel method-based approaches in terms of accuracy. Increasing the component number may improve `Ber`'s performance, but at the cost of higher computational time. Our proposed method was the fastest, achieving a speed-up of several hundred times compared to the SOTA `Bonnet`. Note that `Ber` exhibited unstable behavior in some trials, where $\Delta^2$ exceeded 1000. Such outlier samples were excluded from the results in Table 2.

Our proposed method, based on a linear Hawkes process, as well as the approach based on a nonlinear one (Bonnet & Sangnier, 2025), can estimate inhibitory effects, i.e., triggering kernels that take negative values (see the estimated triggering kernels on the diagonal of Figure 2). However, when using the estimated triggering kernels to predict intensity function $\hat{\lambda}(\cdot)$, our method requires a post-hoc clipping, such as applying $\max(\hat{\lambda}(\cdot), \epsilon)$ for a small $\epsilon$ to ensure the non-negativity of the intensity function. Note that the computational overhead of this post-processing is negligible.

### 4.1.3 EFFECTS OF SUPPORT WINDOW ON PERFORMANCE

The support window $A$ can be regarded as a hyperparameter that controls the shape of the triggering kernel and can therefore be selected from data, for example, via cross-validation. In general, if $A$ is too small, estimation methods cannot capture the true shape of the underlying triggering kernel. Conversely, if $A$ is too large, non-negligible values remain in regions where the true triggering kernel is essentially zero, thereby increasing the estimation error. It is worth noting that choosing $A$ too small is substantially more detrimental than choosing $A$ too large; therefore, it is preferable to set $A$ on the larger side.

To verify this behavior, we evaluated the predictive performance of the proposed method `Ours` for the support window $A \in \{1, 2, 5, 10, 20\}$, using data from the mutually-exciting scenario ($T = 5000$). Here, predictive performance was assessed using the integrated squared error ($\Delta^2$) defined as follows: $\Delta^2 = \sum_{i \in \mathcal{U}} \sum_{j \in \mathcal{U}} \int_0^{20} |g_{ij}(s) - \hat{g}_{ij}(s)|^2 ds$. Table 3 summarizes the results, suggesting that an optimal value of $A$ exists.

Table 4: Results on financial dataset across 10 trials with standard errors in brackets. $nll$ is the negative log-likelihood on test data. $N$ is the data size per trial. $cpu$ is the CPU time in seconds. The other notations follow Table 1.

| $N$ | Exp | | Gau | | Ber | | Bonnet | | Ours | |
|---|---|---|---|---|---|---|---|---|---|---|
| | $nll$ | $cpu$ | $nll$ | $cpu$ | $nll$ | $cpu$ | $nll$ | $cpu$ | $nll$ | $cpu$ |
| 3319 | 8086 | 38.2 | 2899 | 7.17 | **1487** | 13.3 | **1829** | 550 | **1721** | **3.4** |
| | (720) | (15.8) | (1015) | (0.69) | (1291) | (1.40) | (1147) | (71.3) | (896) | (1.43) |

## 4.2 REAL-WORLD DATA

We conducted an experiment on a financial dataset that is widely used for evaluating Hawkes process models (Du et al., 2016). This dataset contains transaction records of a single stock over one day, with two event types ($U = 2$): "buy" and "sell". The event sequence is further partitioned by timestamps. From the 100 sequences available on the GitHub repository of Zuo et al. (2020)[3], each of which size is 3319, we randomly constructed 10 pairs of sequences; for a pair, one was used for model training, and the other was used to evaluate the negative log-likelihood as the predictive error (the lower, the better). For our proposed method, we applied a post-hoc clipping, $\max(\hat{\lambda}(t), 10^{-2})$ to the estimated intensity function. For the models except Exp, the support window $A$ was set at 3, and the hyperparameters were optimized on the grids of $\gamma \in \{0.01, 0.1, 1.0\}$ and $\beta \in \{0.5, 1.0, 10\}$, based on the negative log-likelihood minimization. All other procedures followed those described in Section 4.1.

Table 4 summarizes the results: compared to the baseline methods, our approach achieves robust performance while remaining computationally efficient.

## 5 CONCLUSION

We have proposed a penalized least squares loss formulation for estimating triggering kernels in multivariate Hawkes processes that reside in an RKHS. We demonstrated that a novel representer theorem holds for the optimization problem and derived a feasible estimator based on kernel methods. We evaluated the proposed estimator on synthetic and real-world data, confirming that it achieved comparable predictive accuracy while being substantially faster than the state-of-the-art kernel method estimator. To the best of our knowledge, this is the first example in the long history of kernel methods where a representer theorem holds under RKHS kernels defined by a system of integral equations. We therefore view our result as unique and potentially impactful for the kernel methods community.

*Limitations:* Our proposed method is based on a linear Hawkes process, which does not guarantee the non-negativity of the intensity function. As a result, when using the estimated triggering kernel to predict future intensity values, post-hoc clippings such as applying $\max(\hat{\lambda}(\cdot), \epsilon)$ for a small $\epsilon$ are required. If it is known a priori that the underlying triggering kernels are excitatory, it is more appropriate to enforce non-negativity directly on the estimated triggering kernels.

The computational complexity of our method scales cubically with the dimensionality $U$ of the Hawkes process, making it less suitable for high-dimensional processes, while it empirically works robustly for moderate (up to a few hundred) dimensions $U$ (see Appendix H.2).

A rigorous theoretical analysis of the learning guarantees is beyond the scope of this study and remains an important direction for future research.

---

[3]https://github.com/SimiaoZuo/Transformer-Hawkes-Process

REPRODUCIBILITY STATEMENT

We provide detailed implementation instructions and reproducibility guidelines in Section 4, and the Python code is provided[1].

THE USE OF LARGE LANGUAGE MODELS (LLMS)

We used LLMs to polish writing and correct typos.

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

## A EXAMPLES OF THE ESTIMATED TRIGGERING KERNELS ON SYNTHETIC DATA

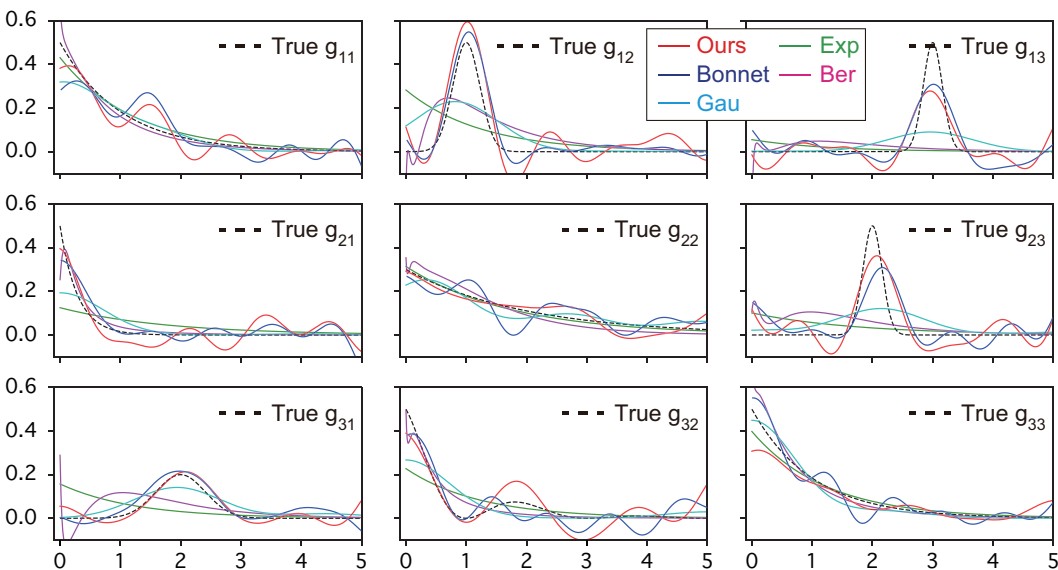

Figure 1: Examples of the estimated triggering kernels in the mutually-exciting scenario. Dashed lines represent the true triggering kernels.

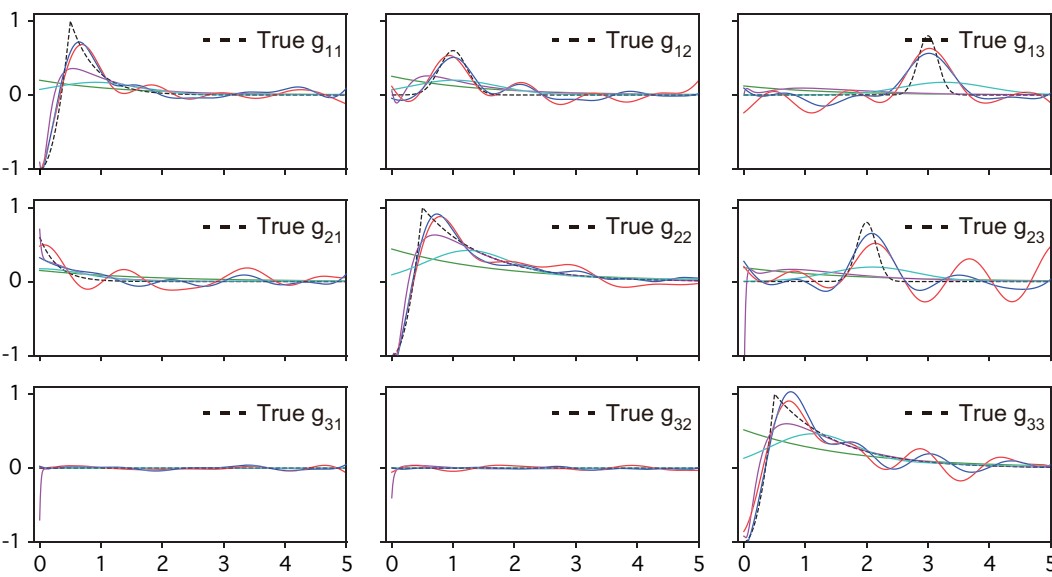

Figure 2: Examples of the estimated triggering kernels in the refractory scenario. Dashed lines represent the true triggering kernels.

## B   PROOF OF THEOREM 1 VIA MERCER'S THEOREM

*Proof.* Through Mercer's theorem, the RKHS kernel $k(\cdot, \cdot)$ can be expressed through its Mercer's expansion:

$$k(t, s) = \sum_{m=1}^{\infty} e_m(t) e_m(s), \quad \int_{\mathcal{T}} e_m(t) e_{m'}(t) dt = \eta_m \delta_{mm'},$$

where $\{e_m(\cdot)\}_{m=1}^{\infty}$ and $\{\eta_m\}_{m=1}^{\infty}$ denote the eigenfunctions and the eigenvalues, respectively, of the integral operator $\int_{\mathcal{T}} \cdot k(t, s) ds$. Accordingly, the triggering kernels in the RKHS, $\{g_{ij}(\cdot) \in \mathcal{H}_k\}_{(i,j) \in \mathcal{U}^2}$, and their squared RKHS norms, $||g_{ij}||_{\mathcal{H}_k}^2$, admit the representation

$$g_{ij}(s) = \sum_{m=1}^{\infty} b_{ij}^m e_m(s), \qquad ||g_{ij}||_{\mathcal{H}_k}^2 = \sum_{m=1}^{\infty} (b_{ij}^m)^2, \qquad (i, j) \in \mathcal{U}^2,$$

where $b = \{b_{ij}^m \in \mathbb{R}\}$ is the expansion coefficient. Using this representation, the optimization problem (3-4) can be reformulated as follows:

$$\hat{b}, \ \hat{\mu} = \arg\min_{b, \ \mu} \left[ L_{\mathrm{LS}}(b, \mu) + \frac{1}{\gamma} \sum_{(i,j) \in \mathcal{U}^2} \sum_{m=1}^{\infty} (b_{ij}^m)^2 \right],$$

where

$$L_{\mathrm{LS}}(b, \mu) = \sum_{i \in \mathcal{U}} \left[ \int_0^T \left( \mu_i + \sum_{n \in \mathcal{N}} \sum_{m=1}^{\infty} b_{iu_n}^m e_m(t - t_n) \mathbf{1}_{0 < t - t_n \le A} \right)^2 dt \right.$$

$$\left. - 2 \sum_{n' \in \mathcal{N}_i} \left( \mu_i + \sum_{n \in \mathcal{N}} \sum_{m=1}^{\infty} b_{iu_n}^m e_m(t_{n'} - t_n) \mathbf{1}_{0 < t_{n'} - t_n \le A} \right) \right].$$

Given the estimate of the baseline intensity $\hat{\mu}$, the optimal coefficient vector $\hat{b}$ must satisfy the equation obtained by setting the gradient of the objective with respect to $b$ equal to zero:

$$\frac{\partial}{\partial b_{ij}^m} \left[ L_{\mathrm{LS}}(b, \hat{\mu}) + \frac{1}{\gamma} \sum_{(i,j) \in \mathcal{U}^2} \sum_{m'=1}^{\infty} (b_{ij}^{m'})^2 \right]\Big|_{b=\hat{b}} = 0, \quad (i, j) \in \mathcal{U}^2, \ m \in \{1, 2, \dots\}. \quad (27)$$

Equation (27) can be written explicitly as

$$\frac{1}{2} \frac{\partial}{\partial b_{ij}^m} \left[ L_{\mathrm{LS}}(b, \hat{\mu}) + \frac{1}{\gamma} \sum_{(i,j) \in \mathcal{U}^2} \sum_{m'=1}^{\infty} (b_{ij}^{m'})^2 \right]\Big|_{b=\hat{b}}$$

$$= \int_0^T \left( \hat{\mu}_i + \sum_{n' \in \mathcal{N}} \sum_{m'=1}^{\infty} \hat{b}_{iu_{n'}}^{m'} e_{m'}(t - t_{n'}) \mathbf{1}_{0 < t - t_{n'} \le A} \right) \sum_{n \in \mathcal{N}_j} e_m(t - t_n) \mathbf{1}_{0 < t - t_n \le A} dt \quad (28)$$

$$- \sum_{n' \in \mathcal{N}_i} \sum_{n \in \mathcal{N}_j} e_m(t_{n'} - t_n) \mathbf{1}_{0 < t_{n'} - t_n \le A} + \frac{1}{\gamma} \hat{b}_{ij}^m = 0.$$

Operating $\sum_{m=1}^{\infty} \cdot e_m(s)$ on both sides of Equation (28) yields the following system of Fredholm integral equations of the second kind:

$$\hat{\mu}_i \sum_{n \in \mathcal{N}_j} \int_0^T k(s, t - t_n) \mathbf{1}_{0 < t - t_n \le A} dt + \sum_{l \in \mathcal{U}} \int_0^T V_{jl}(s, t) \hat{g}_{il}(t) dt$$

$$- \sum_{n' \in \mathcal{N}_i} \sum_{n \in \mathcal{N}_j} k(s, t_{n'} - t_n) \mathbf{1}_{0 < t_{n'} - t_n \le A} + \frac{1}{\gamma} \hat{g}_{ij}(s) = 0, \quad (29)$$

where we used the relation, $\hat{g}_{ij}(s) = \sum_{m=1}^{\infty} \hat{b}_{ij}^m e_m(s)$, and the kernel trick, $k(t, s) = \sum_{m=1}^{\infty} e_m(t) e_m(s)$; here, $V_{jl}(s, t)$ is defined in Equation (6). Equation (29) indicates that the optimal estimator $\hat{g}_{ij}(\cdot)$ admits a linear representation in terms of a set of transformed kernel functions, $\{h_{ij}(\cdot, \cdot)\}_{(i,j) \in \mathcal{U}^2}$, as

$$\hat{g}_{ij}(s) = \sum_{n \in \mathcal{N}_i} h_{ij}(s, t_n) - \hat{\mu}_i \int_0^T h_{ij}(s, t) dt, \quad s \in \mathcal{T}, \ (i, j) \in \mathcal{U}^2,$$

where the transformed kernel functions are defined by a system of simultaneous Fredholm integral equations,

$$\frac{1}{\gamma}h_{ij}(s,s') + \sum_{l\in\mathcal{U}}\int_0^T V_{jl}(s,t)h_{il}(t,s')dt = \sum_{n\in\mathcal{N}_j} k(s,s'-t_n)\mathbf{1}_{0<s'-t_n\leq A}. \tag{30}$$

Since the coefficients in Equation (30) are independent of the index $i$, its solution $h_{ij}(\cdot,\cdot)$ is also independent of $i$, allowing us to write as $h_{ij}(\cdot,\cdot) = h_j(\cdot,\cdot)$. This completes the proof. ∎

## C    PROOF OF THEOREM 1 VIA PATH INTEGRAL REPRESENTATION

*Proof.* Let $\mathcal{K}\cdot(s) = \int_{\mathcal{T}} \cdot\, k(s,t)dt$ be the integral operator with RKHS kernel $k(\cdot,\cdot)$, and $\mathcal{K}^\dagger\cdot(s) = \int_{\mathcal{T}} \cdot\, k^\dagger(s,t)dt$ be its inverse operator. Then, through the path integral representation (Kim, 2021), the squared RKHS norm term can be represented in a functional form,

$$\sum_{(i,j)\in\mathcal{U}^2}\|g_{ij}\|_{\mathcal{H}_k}^2 = \sum_{(i,j)\in\mathcal{U}^2}\iint_{\mathcal{T}\times\mathcal{T}} k^\dagger(s,t)g_{ij}(s)g_{ij}(t)dsdt.$$

Using the representation, the functional derivatives of the least squares term and the penalization term in (3), with respect to $g_{ij}(\cdot)$, can be written as follows:

$$\frac{\delta L_{\text{LS}}}{\delta g_{ij}(s)} = 2\int_0^T\left(\hat{\mu}_i + \sum_{n'\in\mathcal{N}} g_{iu_{n'}}(t-t_{n'})\mathbf{1}_{0<t-t_{n'}\leq A}\right)\sum_{n\in\mathcal{N}_j}\delta(s-(t-t_n))\mathbf{1}_{0<t-t_n\leq A}dt$$

$$-2\sum_{n'\in\mathcal{N}_i}\sum_{n\in\mathcal{N}_j}\delta(s-(t_{n'}-t_n))\mathbf{1}_{0<t_{n'}-t_n\leq A},$$

$$\frac{\delta}{\delta g_{ij}(s)}\sum_{i,j}\|g_{ij}\|_{\mathcal{H}_k}^2 = \frac{\delta}{\delta g_{ij}(s)}\iint_{\mathcal{T}\times\mathcal{T}} k^\dagger(t,t')g_{ij}(t)g_{ij}(t')dtdt' = 2\int_{\mathcal{T}} k^\dagger(s,t)g_{ij}(t)dt,$$

where $\delta(\cdot)$ denotes the Dirac delta function. The optimal estimator $\hat{g}_{ij}(\cdot)$ should solve the equation where the functional derivative of the penalized least squares contrast is equal to zero:

$$\frac{\delta}{\delta g_{ij}(s)}\left[L_{\text{LS}}(g,\hat{\mu}) + \frac{1}{\gamma}\sum_{i,j}\|g_{ij}\|_{\mathcal{H}_k}^2\right]\Bigg|_{g=\hat{g}} = 0, \qquad s\in\mathcal{T},\ (i,j)\in\mathcal{U}^2.$$

Then applying operator $\mathcal{K}$ to the equation leads to the following simultaneous Fredholm integral equations of the second kind:

$$\frac{1}{\gamma}\hat{g}_{ij}(s) + \sum_{l\in\mathcal{U}}\int_0^T V_{jl}(s,t)\hat{g}_{il}(t)dt$$

$$= \sum_{n'\in\mathcal{N}_i}\sum_{n\in\mathcal{N}_j} k(s,t_{n'}-t_n)\mathbf{1}_{0<t_{n'}-t_n\leq A} - \hat{\mu}_i\sum_{n\in\mathcal{N}_j}\int_0^T k(s,t-t_n)\mathbf{1}_{0<t-t_n\leq A}dt, \tag{31}$$

where $V_{jl}(s,t)$ is defined in (6), the second term on the left-hand side of Equation (31) is derived using the following relation,

$$\sum_{n'\in\mathcal{N}}\sum_{n\in\mathcal{N}_j}\int_0^T \hat{g}_{iu_{n'}}(t-t_{n'})k(s,t-t_n)\mathbf{1}_{0<t-t_n\leq A}\mathbf{1}_{0<t-t_{n'}\leq A}dt$$

$$= \sum_{l\in\mathcal{U}}\sum_{n\in\mathcal{N}_j}\sum_{n'\in\mathcal{N}_l}\int_{-t_{n'}}^{T-t_{n'}} \hat{g}_{il}(t)k(s,t+t_{n'}-t_n)\mathbf{1}_{0<t-t_n+t_{n'}\leq A}\mathbf{1}_{0<t\leq A}dt \qquad (t-t_{n'}\to t)$$

$$= \sum_{l\in\mathcal{U}}\int_0^T \hat{g}_{il}(t)\sum_{n\in\mathcal{N}_j}\sum_{n'\in\mathcal{N}_l} k(s,t+t_{n'}-t_n)\mathbf{1}_{\max(t_n,t_{n'})<t+t_{n'}\leq\min(T,A+t_n,A+t_{n'})}\,dt,$$

and the relation, $(\mathcal{K}\mathcal{K}^\dagger)\cdot(s) = \int_\mathcal{T} \cdot \delta(s-t)dt$, was used. Equation (31) indicates that the optimal estimator $\hat{g}_{ij}(\cdot)$ admits a linear representation in terms of a set of transformed kernel functions, $\{h_{ij}(\cdot, \cdot)\}_{(i,j)\in\mathcal{U}^2}$, as

$$\hat{g}_{ij}(s) = \sum_{n\in\mathcal{N}_i} h_{ij}(s, t_n) - \hat{\mu}_i \int_0^T h_{ij}(s,t)dt, \qquad s \in \mathcal{T}, \ (i,j) \in \mathcal{U}^2,$$

where the transformed kernel functions are defined by a system of simultaneous Fredholm integral equations,

$$\frac{1}{\gamma}h_{ij}(s,s') + \sum_{l\in\mathcal{U}} \int_0^T V_{jl}(s,t)h_{il}(t,s')dt = \sum_{n\in\mathcal{N}_j} k(s, s'-t_n)\mathbf{1}_{0<s'-t_n\leq A}. \tag{32}$$

Since the coefficients in Equation (32) are independent of the index $i$, its solution $h_{ij}(\cdot, \cdot)$ is also independent of $i$, allowing us to write as $h_{ij}(\cdot, \cdot) = h_j(\cdot, \cdot)$. This completes the proof. $\blacksquare$

## D  PROOF OF PROPOSITION 2

*Proof.* Given the estimated baseline intensities $\{\hat{\mu}_i\}_{i\in\mathcal{U}}$, the optimal estimators of the triggering kernels, $\{\hat{g}_{ij}(\cdot)\}_{(i,j)\in\mathcal{U}^2}$, are determined by the equations,

$$\frac{\delta}{\delta g_{ij}(s)}\left[L_{\mathrm{LS}}(g,\hat{\mu}) + \frac{1}{\gamma}\sum_{(i,j)\in\mathcal{U}^2} \|g_{ij}\|_{\mathcal{H}_k}^2\right]\Bigg|_{g=\hat{g}} = 0, \qquad s \in \mathcal{T}, \ (i,j) \in \mathcal{U}^2. \tag{33}$$

These equations are equivalently expressed as:

$$\int_0^T \left(\hat{\mu}_i + \sum_{n'\in\mathcal{N}} \hat{g}_{iu_{n'}}(t-t_{n'})\mathbf{1}_{0<t-t_{n'}\leq A}\right) \sum_{n\in\mathcal{N}_j} \delta(s-(t-t_n))\mathbf{1}_{0<t-t_n\leq A}dt$$
$$- \sum_{n'\in\mathcal{N}_i}\sum_{n\in\mathcal{N}_j} \delta(s-(t_{n'}-t_n))\mathbf{1}_{0<t_{n'}-t_n\leq A} + \frac{1}{\gamma}\int_\mathcal{T} k^\dagger(s,t)\hat{g}_{ij}(t)dt = 0. \tag{34}$$

Applying the operator $\int_\mathcal{T} \cdot \hat{g}_{ij}(s)ds$ to both sides of Equations (34) yields the following representation of the RKHS regularization term under the estimated triggering kernels:

$$\frac{1}{\gamma}\|\hat{g}_{ij}\|_{\mathcal{H}_k}^2 = \frac{1}{\gamma}\iint_{\mathcal{T}\times\mathcal{T}} k^\dagger(s,t)\hat{g}_{ij}(s)\hat{g}_{ij}(t)dsdt$$
$$= \sum_{n'\in\mathcal{N}_i}\sum_{n\in\mathcal{N}_j} \hat{g}_{ij}(t_{n'}-t_n)\mathbf{1}_{0<t_{n'}-t_n\leq A} \tag{35}$$
$$- \int_0^T \left(\hat{\mu}_i + \sum_{n'\in\mathcal{N}} \hat{g}_{iu_{n'}}(t-t_{n'})\mathbf{1}_{0<t-t_{n'}\leq A}\right) \sum_{n\in\mathcal{N}_j} \hat{g}_{ij}(t-t_n)\mathbf{1}_{0<t-t_n\leq A}dt.$$

Substituting this representation into the objective function in (3) leads to:

$$L_{\mathrm{LS}}(\hat{g},\hat{\mu}) + \frac{1}{\gamma}\sum_{(i,j)\in\mathcal{U}^2} \|\hat{g}_{ij}\|_{\mathcal{H}_k}^2$$
$$= \sum_{i\in\mathcal{U}}\Bigg[\int_0^T \left(\hat{\mu}_i + \sum_{n\in\mathcal{N}} \hat{g}_{iu_n}(t-t_n)\mathbf{1}_{0<t-t_n\leq A}\right)^2 dt + \sum_{n'\in\mathcal{N}_i}\sum_{n\in\mathcal{N}} \hat{g}_{iu_n}(t_{n'}-t_n)\mathbf{1}_{0<t_{n'}-t_n\leq A}$$
$$- 2\sum_{n'\in\mathcal{N}_i}\left(\hat{\mu}_i + \sum_{n\in\mathcal{N}} \hat{g}_{iu_n}(t_{n'}-t_n)\mathbf{1}_{0<t_{n'}-t_n\leq A}\right)$$
$$- \int_0^T \left(\hat{\mu}_i + \sum_{n'\in\mathcal{N}} \hat{g}_{iu_{n'}}(t-t_{n'})\mathbf{1}_{0<t-t_{n'}\leq A}\right) \sum_{n\in\mathcal{N}} \hat{g}_{iu_n}(t-t_n)\mathbf{1}_{0<t-t_n\leq A}dt\Bigg]$$
$$= \sum_{i\in\mathcal{U}}\Bigg[T\hat{\mu}_i^2 + \hat{\mu}_i \int_0^T \sum_{n\in\mathcal{N}} \hat{g}_{iu_n}(t-t_n)\mathbf{1}_{0<t-t_n\leq A}dt - 2|\mathcal{N}_i|\hat{\mu}_i$$
$$- \sum_{n'\in\mathcal{N}_i}\sum_{n\in\mathcal{N}} \hat{g}_{iu_n}(t_{n'}-t_n)\mathbf{1}_{0<t_{n'}-t_n\leq A}\Bigg]. \tag{36}$$

By invoking the representer theorem (5), the objective can be rewritten as a function of the estimated baseline intensities:

$$
\begin{aligned}
Z(\hat{\mu}) &= L_{\text{LS}}(\hat{g}(\hat{\mu}), \hat{\mu}) + \frac{1}{\gamma} \sum_{(i,j) \in \mathcal{U}^2} \|\hat{g}_{ij}(\hat{\mu})\|_{\mathcal{H}_k}^2 \\
&= \sum_{i \in \mathcal{U}} \Bigg[ \hat{\mu}_i^2 \Big( T - \sum_{n \in \mathcal{N}} \int_0^T \int_0^T h_{u_n}(t - t_n, s) \mathbf{1}_{0 < t - t_n \le A} dt ds \Big) \\
&\quad - \hat{\mu}_i \Big( 2|\mathcal{N}_i| - \sum_{n \in \mathcal{N}} \sum_{n' \in \mathcal{N}_i} \int_0^T h_{u_n}(t - t_n, t_{n'}) \mathbf{1}_{0 < t - t_n \le A} dt \\
&\quad - \sum_{n \in \mathcal{N}} \sum_{n' \in \mathcal{N}_i} \int_0^T h_{u_n}(t_{n'} - t_n, t) \mathbf{1}_{0 < t_{n'} - t_n \le A} dt \Big) \Bigg] + C,
\end{aligned}
\tag{37}
$$

where $C$ is the constant term. $Z(\hat{\mu})$ can be more simplified as

$$
\begin{aligned}
Z(\hat{\mu}) &= \sum_{i \in \mathcal{U}} \Bigg[ \hat{\mu}_i^2 \Big( T - \sum_{n \in \mathcal{N}} \int_0^T \int_0^T h_{u_n}(t - t_n, s) \mathbf{1}_{0 < t - t_n \le A} dt ds \Big) \\
&\quad - 2\hat{\mu}_i \Big( |\mathcal{N}_i| - \sum_{n \in \mathcal{N}} \sum_{n' \in \mathcal{N}_i} \int_0^T h_{u_n}(t - t_n, t_{n'}) \mathbf{1}_{0 < t - t_n \le A} dt \Big) \Bigg] + C,
\end{aligned}
\tag{38}
$$

where the identity

$$
\begin{aligned}
\sum_{n \in \mathcal{N}} \sum_{n' \in \mathcal{N}_i} \int_0^T h_{u_n}(t - t_n, t_{n'}) \mathbf{1}_{0 < t - t_n \le A} dt \\
= \sum_{n \in \mathcal{N}} \sum_{n' \in \mathcal{N}_i} \int_0^T h_{u_n}(t_{n'} - t_n, t) \mathbf{1}_{0 < t_{n'} - t_n \le A} dt,
\end{aligned}
\tag{39}
$$

is used (for proof, see Appendix F). Finally, the optimal estimators $\hat{\mu}_i$ should solve the equation where the functional derivative of $Z(\hat{\mu})$ regarding $\mu_i$ is equal to zero ($dZ/d\mu_i = 0$), which leads to the following representation in terms of the equivalent kernels:

$$
\hat{\mu}_i = \frac{|\mathcal{N}_i| - \sum_{n \in \mathcal{N}} \sum_{n' \in \mathcal{N}_i} \int_0^T h_{u_n}(t - t_n, t_{n'}) \mathbf{1}_{0 < t - t_n \le A} dt}{T - \sum_{n \in \mathcal{N}} \int_0^T \int_0^T h_{u_n}(t - t_n, s) \mathbf{1}_{0 < t - t_n \le A} dt ds}, \qquad i \in \mathcal{U}.
\tag{40}
$$

This completes the proof. ∎

## E   PROOF OF PROPOSITION 3

*Proof.* For an RKHS kernel $k(\cdot, \cdot)$ with the degenerate form given in Equation (13), the coefficient functions, $\{V_{jl}(s, t)\}_{(j,l) \in \mathcal{U}^2}$, appearing in the system of integral equations (6) can also be expressed in degenerate forms as follows:

$$
\begin{aligned}
V_{jl}(s, t) &= \sum_{m=1}^M \phi_m(s) \psi_m^{jl}(t), \\
\psi_m^{jl}(t) &= \sum_{n \in \mathcal{N}_j} \sum_{n' \in \mathcal{N}_l} \phi_m(t + t_{n'} - t_n) \mathbf{1}_{\max(t_n, t_{n'}) < t + t_{n'} \le \min(T, A + t_n, A + t_{n'})}.
\end{aligned}
\tag{41}
$$

Substituting Equation (41) into Equation (6), we find that the solutions, $\{h_j(s, s')\}_{j \in \mathcal{U}}$, admit degenerate forms as

$$
\begin{aligned}
h_j(s, s') &= \gamma \sum_{n \in \mathcal{N}_j} k(s, s' - t_n) \mathbf{1}_{0 < s' - t_n \leq A} - \gamma \sum_{l \in \mathcal{U}} \int_0^T V_{jl}(s, t) h_l(t, s') dt, \\
&= \gamma \sum_{m=1}^M \phi_m(s) \left[ \sum_{n \in \mathcal{N}_j} \phi_m(s' - t_n) \mathbf{1}_{0 < s' - t_n \leq A} - \sum_{l \in \mathcal{U}} \int_0^T \psi_m^{jl}(t) h_l(t, s') dt \right] \quad (42) \\
&= \sum_{m=1}^M \phi_m(s) c_m^j(s'),
\end{aligned}
$$

where $\{c_m^j(s')\}_{(m,j) \in [\![1,M]\!] \times \mathcal{U}}$ are unknown coefficient functions. By substituting Equations (42) and (41) into Equation (6), we obtain the following linear system that the coefficient functions must satisfy:

$$
\sum_m \phi_m(s) \left[ \frac{1}{\gamma} c_m^j(s') + \sum_{l \in \mathcal{U}} \int_0^T \psi_m^{jl}(t) \sum_{m'} \phi_{m'}(t) c_{m'}^l(s') dt - \sum_{n \in \mathcal{N}_j} \phi_m(s' - t_n) \mathbf{1}_{0 < s' - t_n \leq A} \right] = 0,
$$

$$
\therefore \frac{1}{\gamma} c_m^j(s) + \sum_{l \in \mathcal{U}} \sum_{m'} c_{m'}^l(s) \int_0^T \psi_m^{jl}(t) \phi_{m'}(t) dt - \sum_{n \in \mathcal{N}_j} \phi_m(s - t_n) \mathbf{1}_{0 < s - t_n \leq A} = 0, \quad (43)
$$

for $(m, j) \in [\![1, M]\!] \times \mathcal{U}$. Let us define the $MU$-dimensional stacked vector of coefficient functions as

$$
\tilde{c}(s) = (c_1^1(s), c_2^1(s), \ldots, c_M^1(s), c_1^2(s), c_2^2(s), \ldots, c_M^2(s), c_1^3(s), \ldots, c_M^U(s))^\top. \quad (44)
$$

Then, the linear system can be written compactly as

$$
\left( \frac{1}{\gamma} \boldsymbol{I}_{MU} + \boldsymbol{\Xi} \right) \tilde{c}(s) = \tilde{\phi}(s), \quad (45)
$$

where $\tilde{\phi}(s)$ and $\boldsymbol{\Xi}$ are defined in Equations (15) and (16), respectively. Substituting Equation (45) into Equation (42) yields the solution to the system of integral equations (6) as,

$$
\begin{aligned}
h_j(s, s') &= \sum_{m=1}^M \phi_m(s) c_m^j(s') \\
&= \phi(s)^\top \left[ c(s') \right]_{1 + (j-1)M : jM} \quad (46) \\
&= \phi(s)^\top \left[ \left( \frac{1}{\gamma} \boldsymbol{I}_{MU} + \boldsymbol{\Xi} \right)^{-1} \tilde{\phi}(s') \right]_{1 + (j-1)M : jM}.
\end{aligned}
$$

This completes the proof. ∎

## F    Proof of Proposition 4

*Proof.* Substituting Equation (14) into Equation (5) yields the expression for the estimated triggering kernels in terms of the feature maps:

$$
\hat{g}_{ij}(s) = \phi(s)^\top \left[ \left( \frac{1}{\gamma} \boldsymbol{I}_{MU} + \boldsymbol{\Xi} \right)^{-1} \left( \sum_{n \in \mathcal{N}_i} \tilde{\phi}(t_n) - \hat{\mu}_i \int_0^T \tilde{\phi}(t) dt \right) \right]_{1 + (j-1)M : jM}. \quad (47)
$$

By using Equation (14), the double integral in the denominator of Equation (12) can be rewritten using the feature maps as follows:

$$\sum_{n \in \mathcal{N}} \int_0^T \int_0^T h_{u_n}(t - t_n, s) \mathbf{1}_{0 < t - t_n \leq A} dt ds$$

$$= \sum_{n \in \mathcal{N}} \left[ \int_0^T \phi(t - t_n) \mathbf{1}_{0 < t - t_n \leq A} dt \right]^\top \left[ \left( \frac{1}{\gamma} \boldsymbol{I}_{MU} + \boldsymbol{\Xi} \right)^{-1} \int_0^T \tilde{\phi}(s) ds \right]_{1 + (u_n - 1)M : u_n M}$$

$$= \sum_{l \in \mathcal{U}} \left[ \sum_{n \in \mathcal{N}_l} \int_0^T \phi(t - t_n) \mathbf{1}_{0 < t - t_n \leq A} dt \right]^\top \left[ \left( \frac{1}{\gamma} \boldsymbol{I}_{MU} + \boldsymbol{\Xi} \right)^{-1} \int_0^T \tilde{\phi}(s) ds \right]_{1 + (l - 1)M : lM}$$

$$= \sum_{l \in \mathcal{U}} \left[ \int_0^T \tilde{\phi}_l(t) dt \right]^\top \left[ \left( \frac{1}{\gamma} \boldsymbol{I}_{MU} + \boldsymbol{\Xi} \right)^{-1} \int_0^T \tilde{\phi}(s) ds \right]_{1 + (l - 1)M : lM}$$

$$= \left( \int_0^T \tilde{\phi}(t) dt \right)^\top \left( \frac{1}{\gamma} \boldsymbol{I}_{MU} + \boldsymbol{\Xi} \right)^{-1} \left( \int_0^T \tilde{\phi}(t) dt \right). \tag{48}$$

Similarly, the integral term in the numerator of Equation (12) becomes,

$$\sum_{n \in \mathcal{N}} \sum_{n' \in \mathcal{N}_i} \int_0^T h_{u_n}(t - t_n, t_{n'}) \mathbf{1}_{0 < t - t_n \leq A} dt$$

$$= \sum_{n \in \mathcal{N}} \left[ \int_0^T \phi(t - t_n) \mathbf{1}_{0 < t - t_n \leq A} dt \right]^\top \left[ \left( \frac{1}{\gamma} \boldsymbol{I}_{MU} + \boldsymbol{\Xi} \right)^{-1} \left( \sum_{n' \in \mathcal{N}_i} \tilde{\phi}(t_{n'}) \right) \right]_{1 + (u_n - 1)M : u_n M}$$

$$= \sum_{l \in \mathcal{U}} \left( \int_0^T \tilde{\phi}_l(t) dt \right)^\top \left[ \left( \frac{1}{\gamma} \boldsymbol{I}_{MU} + \boldsymbol{\Xi} \right)^{-1} \left( \sum_{n' \in \mathcal{N}_i} \tilde{\phi}(t_{n'}) \right) \right]_{1 + (l - 1)M : lM}$$

$$= \left( \int_0^T \tilde{\phi}(t) dt \right)^\top \left( \frac{1}{\gamma} \boldsymbol{I}_{MU} + \boldsymbol{\Xi} \right)^{-1} \left( \sum_{n' \in \mathcal{N}_i} \tilde{\phi}(t_{n'}) dt \right). \tag{49}$$

This completes the proof. ∎

Furthermore, from Equation (49) and the following identity:

$$\sum_{n \in \mathcal{N}} \sum_{n' \in \mathcal{N}_i} \int_0^T h_{u_n}(t_{n'} - t_n, t) \mathbf{1}_{0 < t_{n'} - t_n \leq A} dt$$

$$= \sum_{n \in \mathcal{N}} \left[ \sum_{n' \in \mathcal{N}_i} \phi(t_{n'} - t_n) \mathbf{1}_{0 < t_{n'} - t_n \leq A} \right]^\top \left[ \left( \frac{1}{\gamma} \boldsymbol{I}_{MU} + \boldsymbol{\Xi} \right)^{-1} \left( \int_0^T \tilde{\phi}(t) dt \right) \right]_{1 + (u_n - 1)M : u_n M}$$

$$= \sum_{l \in \mathcal{U}} \left[ \sum_{n' \in \mathcal{N}_i} \sum_{n \in \mathcal{N}_l} \phi(t_{n'} - t_n) \mathbf{1}_{0 < t_{n'} - t_n \leq A} \right]^\top \left[ \left( \frac{1}{\gamma} \boldsymbol{I}_{MU} + \boldsymbol{\Xi} \right)^{-1} \left( \int_0^T \tilde{\phi}(t) dt \right) \right]_{1 + (l - 1)M : lM}$$

$$= \sum_{l \in \mathcal{U}} \left[ \sum_{n' \in \mathcal{N}_i} \tilde{\phi}_l(t_{n'}) \right]^\top \left[ \left( \frac{1}{\gamma} \boldsymbol{I}_{MU} + \boldsymbol{\Xi} \right)^{-1} \left( \int_0^T \tilde{\phi}(t) dt \right) \right]_{1 + (l - 1)M : lM}$$

$$= \left( \sum_{n' \in \mathcal{N}_i} \tilde{\phi}(t_{n'}) dt \right)^\top \left( \frac{1}{\gamma} \boldsymbol{I}_{MU} + \boldsymbol{\Xi} \right)^{-1} \left( \int_0^T \tilde{\phi}(t) dt \right)$$

$$= \left( \int_0^T \tilde{\phi}(t) dt \right)^\top \left( \frac{1}{\gamma} \boldsymbol{I}_{MU} + \boldsymbol{\Xi} \right)^{-1} \left( \sum_{n' \in \mathcal{N}_i} \tilde{\phi}(t_{n'}) dt \right),$$

where the final equality holds because $\left( \gamma^{-1} \boldsymbol{I}_{MU} + \boldsymbol{\Xi} \right)$ is symmetric, we obtain the relation in Equation (39), which holds for any $M \leq \infty$ and feature map $\phi(s)$.

Table 5: Average CPU time in seconds across 10 trials. $\tilde{N}$ denotes the average data size per trial.

| | | Bonnet | Ours |
|---|---|---|---|
| $T$ | $\tilde{N}$ | $cpu$ | $cpu$ |
| 10000 | 8248 | 9250 | 19.1 |
| 15000 | 12748 | 26406 | 29.5 |

## G   EXISTENCE OF EQUIVALENT KERNELS

We demonstrate that the equivalent kernels, $\{h_j(\cdot, \cdot)\}_{j \in \mathcal{U}}$, defined by the system of Fredholm integral equations (6) exist and are uniquely determined for any positive semi-definite kernel $k(\cdot, \cdot)$.

A positive semi-definite kernel can be represented as the inner product of an $M$-dimensional feature vector, $\boldsymbol{\phi}(\cdot) = (\phi_1(\cdot), \dots, \phi_M(\cdot))^\top$, for some $M \leq \infty$. Under this representation, the equivalent kernels, $\{h_j(\cdot, \cdot)\}_{j \in \mathcal{U}}$, can be expressed by Equation (14). The block matrix $\boldsymbol{\Xi} = [\boldsymbol{\Xi}_{ij}] \in \mathbb{R}^{MU \times MU}$ in Equation (14) has a set of submatrices defined in Equation (15), which can be rewritten as follows:

$$
\begin{aligned}
\boldsymbol{\Xi}_{ij} &= \sum_{n \in \mathcal{N}_i} \sum_{n' \in \mathcal{N}_j} \mathbf{1}_{\max(t_n, t_{n'}) < \min(T, A+t_n, A+t_{n'})} \int_{\max(t_n, t_{n'})}^{\min(T, A+t_n, A+t_{n'})} \boldsymbol{\phi}(t - t_n) \boldsymbol{\phi}(t - t_{n'})^\top dt \\
&= \sum_{n \in \mathcal{N}_i} \sum_{n' \in \mathcal{N}_j} \int_0^T \xi_n(t) \xi_{n'}(t) \boldsymbol{\phi}(t - t_n) \boldsymbol{\phi}(t - t_{n'})^\top dt,
\end{aligned} \tag{50}
$$

where $\xi_n(t) = \mathbf{1}_{t_n < t < \min(T, A+t_n)}$. Therefore, for all $\boldsymbol{c} = (c_{11}, \dots, c_{1M}, c_{21}, \dots, c_{UM})^\top \in \mathbb{R}^{MU}$, the following inequality holds:

$$
\begin{aligned}
&\boldsymbol{c}^\top \boldsymbol{\Xi} \boldsymbol{c} \\
&= \sum_{i \in \mathcal{U}} \sum_{j \in \mathcal{U}} \sum_{m=1}^M \sum_{m'=1}^M c_{im} c_{jm'} \sum_{n \in \mathcal{N}_i} \sum_{n' \in \mathcal{N}_j} \int_0^T \xi_n(t) \xi_{n'}(t) \phi_m(t - t_n) \phi_{m'}(t - t_{n'}) dt, \\
&= \int_0^T \left[ \sum_{i \in \mathcal{U}} \sum_{m=1}^M c_{im} \sum_{n \in \mathcal{N}_i} \xi_n(t) \phi_m(t - t_n) \right] \left[ \sum_{j \in \mathcal{U}} \sum_{m'=1}^M c_{jm'} \sum_{n' \in \mathcal{N}_j} \xi_{n'}(t) \phi_{m'}(t - t_{n'}) \right] dt \\
&= \int_0^T \left[ \sum_{i \in \mathcal{U}} \sum_{m=1}^M c_{im} \sum_{n \in \mathcal{N}_i} \xi_n(t) \phi_m(t - t_n) \right]^2 dt \\
&\geq 0.
\end{aligned} \tag{51}
$$

This relation shows that $\left( \frac{1}{\gamma} \boldsymbol{I}_{MU} + \boldsymbol{\Xi} \right)$ in Equation (14) is positive definite and invertible for $\gamma > 0$. Therefore, the equivalent kernels defined by Equations (6) and (14) exist and are uniquely determined for any positive semi-definite kernels $k(\cdot, \cdot)$.

## H   ADDITIONAL EXPERIMENTS

### H.1   SCALABILITY ON LARGER DATA SIZE

In Section 2.4, we discussed the scalability of the proposed method on the data size. To confirm it, we conducted an experiment in the refractory scenario with $T \in \{10000, 15000\}$, and evaluated the CPU times of Ours and Bonnet on these larger datasets. The results in Table 5 demonstrate that Ours remains scalable for the larger data sizes.

### H.2   SCALABILITY ON LARGER DIMENSIONALITY

The computational cost of our method (Ours) scales cubically with the dimensionality $U$, which is a disadvantage compared to the quadratic scaling of the prior kernel method (Bonnet). We

Table 6: Average CPU time in seconds across 5 trials.

| $U$ | Exp
*cpu* | Gau
*cpu* | Ber
*cpu* | Bonnet
*cpu* | Ours
*cpu* |
|---|---|---|---|---|---|
| 3 | 124 | 7.03 | 10.8 | 413 | 5.04 |
| 15 | 1369 | 300 | 254 | 8513 | 29.9 |

Table 7: CPU time in seconds of `Ours` regarding the dimensionality $U$ across 5 trials with standard errors in brackets.

| $U = 10$
*cpu* | $U = 50$
*cpu* | $U = 100$
*cpu* | $U = 200$
*cpu* | $U = 300$
*cpu* | $U = 500$
*cpu* |
|---|---|---|---|---|---|
| 0.68 | 5.05 | 26.9 | 160 | 478 | 2010 |
| (0.26) | (0.82) | (0.67) | (1.91) | (6.70) | (15.5) |

conducted an experiment under a refractory scenario with $T = 2000$ and $U = 15$ to examine this issue. The triggering kernel matrix was constructed as a $U \times U$ block-diagonal matrix obtained by placing copies of the $3 \times 3$ triggering kernel matrix, $g(s) = [g_{ij}(s)]_{ij}$, used in Section 4.1.2, along the diagonal. We fixed the hyperparameters to $(\gamma, \beta) = (1, 1)$ and evaluated only the computation time.

The results in Table 6 (note that the $U = 3$ case is identical to that reported in Table 2 for $T = 2000$) show that all methods exhibited an increase in computation time as $U$ grows. However, the increase for `Ours` is more moderate compared to the conventional methods (`Exp`, `Gau`, `Ber`, and `Bonnet`). Although this trend may contradict the complexity analysis presented in Section 2.4, it can be attributed to the fact that our method relies solely on matrix additions and matrix inversions (performed via Cholesky decomposition), which are highly amenable to parallelization across multiple CPU cores.

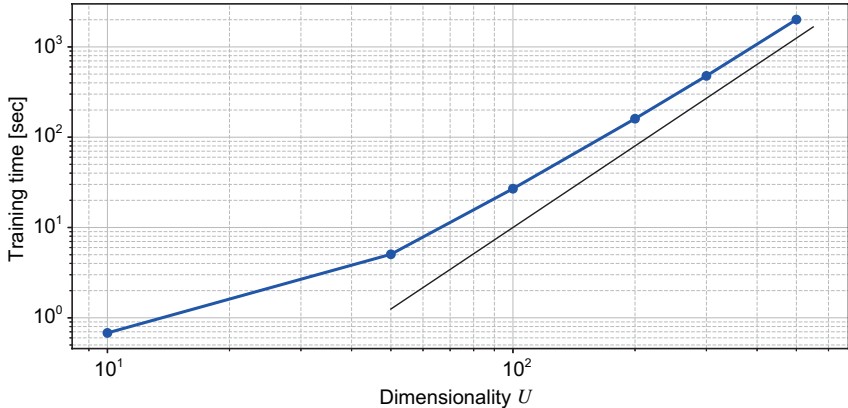

Figure 3: Scaling curve of the training time regarding the dimensionality. The blue line represents the training time of `Ours` with respect to the dimensionality $U$. The black line represents a reference curve of $\mathcal{O}(U^3)$.

We further conducted an experiment to empirically characterize how the computational cost of our method scales with respect to $U$. The computational burden of solving the linear system is most evident in settings where the number of events per dimension is small while $U$ itself is large. Accordingly, we set $\mu_i = 0.1$, $g_{ij}(\cdot) = 0$, and $T = 100$, and generated synthetic datasets for $U \in \{10, 50, 100, 200, 300, 500\}$. We then measured the training time of the proposed method. Table 7 and Figure 3 summarize the results: empirically, the training time grows sub-cubically in $U$ up to around $U \simeq 100$, and the scaling curve gradually approaches $\mathcal{O}(U^3)$. At least up to $U = 500$, the observed scaling is therefore better than $\mathcal{O}(U^3)$. We attribute this behavior largely to the highly

Table 8: Estimation accuracy of triggering kernels and predictive performance by `Ours`. Results are on the mutually exciting scenario dataset, averaged over 10 trials, with standard errors in brackets. $nll$ is the negative log-likelihood on the test data. $\Delta^2$ denotes the integrated squared error for triggering kernel estimation, of which results are a reproduction of those presented in Table 1.

| $T = 2000$ | | $T = 3000$ | | $T = 5000$ | |
| $nll$ | $\Delta^2$ | $nll$ | $\Delta^2$ | $nll$ | $\Delta^2$ |
| --- | --- | --- | --- | --- | --- |
| 3294 | 0.38 | 3013 | 0.27 | 2842 | 0.20 |
| (481) | (0.15) | (226) | (0.06) | (194) | (0.16) |

optimized implementation of Cholesky factorization in TensorFlow, which efficiently exploits the 12-core CPU architecture.

Additionally, we implemented a naïve conjugate gradient (CG) solver to compare its runtime with the Cholesky factorization (CF) approach, using a very simple preconditioner given by the inverse of the diagonal of $(\frac{1}{\gamma} I_{MU} + \Xi)$. Unfortunately, this configuration led to worse runtime than CF, suggesting that a substantially more suitable preconditioner is required to fully benefit from CG. Designing such a preconditioner is nontrivial and beyond the scope of this study. Based on these findings, our current model should be regarded as being practically targeted at event data with up to a few hundred dimensions.

### H.3 EFFECTS OF HYPERPARAMETER GRID ON PERFORMANCE

For the proposed model `Ours`, we conducted an experiment under the refractory scenario with $T = 5000$, where the grid was refined from $3 \times 3$ to $10 \times 10$. The resulting squared error $\Delta^2$ was $0.58 \pm 0.12$, which represents only a marginal improvement over the $3 \times 3$ grid ($0.59 \pm 0.13$). This result suggests that the performance in Tables 1-2, especially the gap between `Bonnet` and `Ours`, could not be solely attributed to the hyperparameter tuning strategy. Since `Bonnet` is based on the likelihood function, it is expected to achieve higher accuracy than `Ours`, which relies on the least squares loss. Note that maximum likelihood estimation is known to be statistically efficient asymptotically for Hawkes processes (see (Ogata, 1978)).

### H.4 EFFECTS OF ESTIMATION ACCURACY OF TRIGGERING KERNELS ON PREDICTIVE PERFORMANCE

We conducted an additional experiment to examine the relationship between the estimation accuracy of the latent triggering kernel and the predictive performance of the point process model. Based on the mutually-exciting scenario dataset in Section 4.1.1, we estimated the triggering kernel for each trial of $T \in \{2000, 3000, 5000\}$ by using the proposed model (`Ours`), and evaluated its predictive performance on $T = 7000$ data. Here, predictive performance was assessed using the negative log-likelihood (the lower, the better). Since the $T = 7000$ data is composed of 10 trials, we report the average predictive performance over these trials. In addition, because each training dataset also contains 10 trials, we repeated the above prediction experiment 10 times. We applied a post-hoc clipping, $\max(\hat{\lambda}(t), 10^{-4})$, to the estimated intensity function $\hat{\lambda}(\cdot)$. The results, summarized in Table 8, show that the predictive performance of the point process model indeed improves as the estimation accuracy of the latent triggering kernel increases.

## I DEEP NEURAL NETWORK MODELS VS. KERNEL METHODS

Deep neural network (DNN) models currently dominate much of the machine learning literature. Nevertheless, approaches based on kernel methods remain highly valuable for several important reasons.

**Reliability of optimization:** Kernel method-based models (including our proposed method) typically reduce to convex optimization problems, which enjoy a unique global optimum and stable training behavior. As a result, they are not subject to the reproducibility issues often observed in contemporary DNNs, where the final performance can vary significantly depending on the optimization algorithm, its tuning parameters, and the parameter initialization. Moreover, kernel method-based

models involve only a few regularization parameters and a small number of kernel parameters, resulting in significantly fewer hyperparameters than DNNs. This makes the models substantially easier to tune in practice.

**Model interpretability:** Many DNN-based Hawkes processes (or more generally, history-dependent temporal point processes) embed the past event history into a latent vector via a DNN and directly output the conditional intensity of the next event (Zuo et al., 2020; Du et al., 2016). While this approach is powerful, especially for marked point processes with unstructured marks and complex history dependence, the influence of past events on future events is effectively a black box, making the event-generation dynamics difficult to interpret or explain. In contrast, both conventional kernel method-based models and our proposed method explicitly learn triggering kernels, which provide a transparent description of how past events affect future intensity functions, thereby offering high interpretability.

**Complementarity of kernel methods and deep models:** We also view it as unproductive to position kernel methods and DNNs as fundamentally competing paradigms. Hybrid approaches such as deep kernel learning (Wilson et al., 2016), which combine the computational advantages of kernel methods with the representational power of DNNs, are promising directions for future research. Our current work contributes to this broader agenda by strengthening the theoretical and methodological foundations on the kernel side, which can subsequently be integrated with deep architectures.

