# OpenReview forum: "A Representer Theorem for Hawkes Processes via Penalized Least Squares Minimization"
_ICLR.cc/2026/Conference — ICLR 2026 Oral_

### Official Review · Reviewer_JiiZ · 2025-10-31

**Soundness:** 3
**Presentation:** 3
**Contribution:** 2
**Rating:** 6
**Confidence:** 3

**Summary:**

The authors consider nonparametric estimation of the triggering kernels in a multivariate linear Hawkes process using an RKHS (reproducing kernel Hilbert space) framework. They formulate a penalised least-squares loss for the latent triggering functions, and derive a novel representer theorem showing that the optimal kernel estimators admits an expansion over a family of transformed kernels evaluated at data points, and remarkably, the dual coefficients are analytically fixed to unity, so one does not need to solve an optimization problem. These transformed kernels are defined by a system of Fredholm integral equations, which allows an efficient random-feature approximation algorithm, further enabling scalability since number of random features can be typically set much smaller than number of samples.

The method applies to linear multivariate Hawkes (identity link function, no inhibitory effects or non-linear link). So the generality is narrower than the Bonnet & Sangnier (2025) paper which handles inhibition and non-linear link functions. The trade-off is computational simplicity and analytic coefficient expressions.

Empirical results on synthetic toy-models show competitive accuracy and much better efficiency compared to existing kernel-based methods.

**Strengths:**

(1) Novel representer theorem: While prior works (Bonnet & Sangnier 2025) derived a general representer theorem for approximate loss formulations (and requiring optimization of dual coefficients), this paper focuses on linear Hawkes processes and provides one for the exact penalised least squares loss obtaining that dual coefficients equal unity w.r.t. transformed kernels characterised by Fredholm integral equations  That is a non-trivial theoretical advance.

(2) Practical algorithm: To obtain practical algorithm, authors use random features to avoid discretization approximation of the integral operators,  and, in process, reduce the inversion to a matrix independent of data size. This results in practical algorithm that scales much better than previous kernel-based methods for Hawkes. Further, authors empirically show that you can maintain competitive predictive accuracy while significantly improving computational efficiency. This allows algorithm's deployment in large event-data applications.

(3) Paper is generally well written and related works are, up to my knowledge, well covered.

**Weaknesses:**

(1) Limited depth: The paper is fundamentally methodological. New kernel method is backed by the representer theorem and coupled with random features to result in practical algorithm. However, any form of learning guarantees is lacking.

(2) Scalability via random features: Using random features and reducing the inversion to a matrix independent of data size is golden standard in scaling kernel methods. Hence, there is hardly some big novelty in this. Even more so, since there is lack on any theory such as e.g. convergence w.r.t. number of random features.

(3) Empirical evaluation: Since the paper is methodological,  testing it only on synthetic toy-models is a bit underwhelming.

To conclude, up to my knowledge, the paper delivers a clear meaningful advance: the theoretical representer result is strong (analytic coefficients) and the algorithmic scalability is obtained via random features. As such, it can make a solid contribution to kernel-based nonparametric estimation for Hawkes processes if the empirical study is improved, or some theoretical  learning guarantees included.

**Questions:**

Minor issues:
- line 278 it should read "number"
- line 132, definition of ${\cal N}_i$ is missing, though intuivily clear what it should represent
- line 147, would be good to explicitly write what dual coefficients equal to
- line 155 personally I would prefer using $\dagger$ for pseudo inverse sainted of $*$ (typically adjoint)

---

> ### Author Response · Authors · 2025-11-21
>
> We would like to thank the reviewer for the constructive comments. Below, we provide a detailed response to each of the comments.
>
> **The method applies to linear multivariate Hawkes (identity link function, no inhibitory effects or non-linear link). So the generality is narrower than the Bonnet & Sangnier (2025) paper which handles inhibition and non-linear link functions.**
>
> Actually, our proposed method can capture inhibitory effects, as demonstrated in our experiments on the refractory scenario. Please refer to the estimated trigger kernels on the diagonal of Figure A2, which clearly display inhibitory patterns. Note that the identity link function in our model requires post-hoc clippings, such as applying $\max(\lambda(t), \epsilon)$ for a small $\epsilon$, when using the estimated triggering kernel to predict future intensity values. We acknowledge that our original explanation may have been unclear, and in the revised manuscript, we will explicitly state that our method is capable of estimating inhibitory effects.
>
> **As such, it can make a solid contribution to kernel-based nonparametric estimation for Hawkes processes if the empirical study is improved, or some theoretical learning guarantees included.**
>
> We have added an experiment on a real-world dataset, demonstrating that our approach achieves robust performance while remaining computationally efficient. For further details, please refer to our initial response to Reviewer EXZy.
>
> A rigorous theoretical analysis of the learning guarantees is unfortunately beyond what we can complete within the rebuttal period. We will therefore explicitly acknowledge this point as a limitation of the present work in the revised manuscript and highlight it as an important direction for future research.
>
> We hope these adequately address the reviewer’s concern.
>
> **Minor issues:**
>
> We thank the reviewer for the suggestions and for pointing out the typos. We will reflect them appropriately in the revised manuscript.

---

> > ### Comment · Reviewer_JiiZ · 2025-11-26
> >
> > I thank the authors for their reply and clarifications. As stated in my review, I lean towards the acceptance, and will keep my score.

---

### Official Review · Reviewer_wcyH · 2025-11-01

**Soundness:** 3
**Presentation:** 3
**Contribution:** 3
**Rating:** 8
**Confidence:** 4

**Summary:**

The paper aims to provide a theoretically principled methodology for reducing the computational complexity of nonparametric estimation of the intensity functions in multivariate linear Hawkes process, where the triggering "kernels" $g_{ij}$ are assumed to belong to an RKHS $\mathcal{H}_{k}$ with reproducing kernel $k$ and a Tikhonov regularization is employed.

The novel technical contribution of this paper and the workhorse behind this paper is a representer theorem for the sample based estimates $\hat{g}_{ij}$ in terms of  linear combinations of Fredholm integral transformed versions of the base kernel function $k$, denoted by $h_j$'s. The representer theorem ensures that there is no need for dual variable (coefficients in terms of $h_ij$'s) optimization, leaving only the system of Fredholm integral equations for $h_j$'s to be solved. Using the machinery of Random Fourier Features (RFF), a practical algorithm is proposed with corresponding experiments based on synthetic data that achieves $O(N^{2}M^{2}U^{2} +M^{3}U^{3})$ computational complexity where $N$ is the maximum sampling frequency in each dimension, M is the number of Random features and U is the dimensionality of the multivariate linear Hawkes process, improving upon $O(N^{4}U^{2})$ computational complexity of the SOTA method (Bonnet and Sangnier, 2025), along with the usual memory complexity improvement expected from RFF. While computation time on synthetic experiments drops by around 2 orders of magnitude, inferential accuracy loss is reasonably low.

**Strengths:**

The paper presents a very clear and precise solution to a significant computational problem in the context of linear Hawkes processes,and explains the utility behind analyzing the continuous variational formulation of the data fitting problem, along with a clean and complete derivation of the key technical result being included. The representer theorem is novel in the context of this particular problem, and shows the theoretical advantage of using the Tikhonov regualrization of RKHS norm in contrast to the Hakwes process log-likelihood based estimation approach. Synthetic experiments also show around 100 times relative improvement in computational complexity compared to SOTA with small loss of inferential power for the trigerring "kernels" $g$, in terms of integrated squared error.

**Weaknesses:**

While the RFF approach indeed allows a practical solution to the problem, from a theoretical perspective it would be helpful if the authors can report the specific conditions (properties of kernels etc.) under which the Fredholm integral equations for $h_j$'s (Equation 6 under Theorem 1) admit a solution. The authors themselves acknowledge the limitation of the paper being restricted to linear Hawkes processes and requires adhoc post-processing in empirical experiments, while approach of the SOTA method is more general and is applicable to nonlinear processes as well. Also, from a empirical perspective, the implication of the small loss of inferential power for the latent trigerring "kernels" $g_{ij}$'s for actual predictive performance for the point process is not immediately clear.

Minor typos:
1. Replace $x$'s by $s$ in Equation 22 (Lines 452 and 453)
2. Replace "quantifing" by quantifying on Line 096
3. Replace "evet" by event on Ine 134

**Questions:**

Based on the discussion of the weaknesses, the paper will improve if
1.  the authors can report the specific conditions (properties of kernels etc.) under which the Fredholm integral equations for $h_j$'s (Equation 6 under Theorem 1) admit a solution and
2. demonstrate the implication of the small loss of inferential power for the latent trigerring "kernels" $g_{ij}$'s for actual predictive performance for the point process.

Although this did not play a role in my assessment, but, as always, a real-life experimment will always help, especially regarding the evaluation of predictive performance.

---

> ### Author Response · Authors · 2025-11-21
>
> We would like to thank the reviewer for the highly positive and constructive comments, which have encouraged us greatly. Below, we provide a detailed response to each of the comments.
>
> **the authors can report the specific conditions (properties of kernels etc.) under which the Fredholm integral equations for $h_j$'s (Equation 6 under Theorem 1) admit a solution**
>
> We understand the reviewer's question as asking whether, for certain choices of data or kernel function, the equivalent kernel $h_j (\cdot,\cdot)$ might fail to be well-defined. If this interpretation is correct, we answer that for any positive semi-definite kernel $k(\cdot,\cdot)$, the equivalent kernel $h_j (\cdot,\cdot)$ exists and is uniquely defined. The key idea is that the matrix $\Xi$ appearing in Eq. (11) is positive semi-definite for any feature dimension $M \leq \infty$. Consequently, the matrix $(\frac{1}{\gamma}I_{MU} + \Xi)$ is invertible, which guarantees the existence (and uniqueness) of the equivalent kernel $h_j (\cdot,\cdot)$ for an arbitrary positive semi-definite kernel $k(\cdot,\cdot)$. Since the full derivation is somewhat involved, we have included it in the manuscript as Appendix G (please see the revised rebuttal manuscript).
>
> **demonstrate the implication of the small loss of inferential power for the latent trigerring "kernels" $g_ij$'s for actual predictive performance for the point process.**
>
> Please allow us a few more days to prepare our response.
>
> **Although this did not play a role in my assessment, but, as always, a real-life experimment will always help, especially regarding the evaluation of predictive performance.**
>
> This concern was raised by all reviewers. In response, we have added an experiment on a real-world dataset, demonstrating that our approach achieves robust performance while remaining computationally efficient. For further details, please refer to our first response to Reviewer EXZy.
>
> **Minor typos:**
>
> We thank the reviewer for pointing out the typos. We will fix them.

---

> > ### Author Response · Authors · 2025-11-27
> >
> > **demonstrate the implication of the small loss of inferential power for the latent trigerring "kernels" $g_ij$'s for actual predictive performance for the point process.**
> >
> > We apologize for the delay in submitting our response to this comment and hope that it addresses the reviewer's question.
> >
> > Following the reviewer's suggestion, we conducted an additional experiment to examine the relationship between the estimation accuracy of the latent triggering kernel and the predictive performance of the point process model. Based on the mutually-exciting scenario dataset in Section 4.1, we estimated the triggering kernel for each trial of $T \in \\{2000, 3000, 5000 \\}$ by using the proposed model (Ours), and evaluated its predictive performance on $T = 7000$ data. Here, predictive performance was assessed using the negative log-likelihood (the lower, the better). Since the $T = 7000$ data is composed of 10 trials, we report the average predictive performance over these trials. In addition, because each training dataset also contains 10 trials, we repeated the above prediction experiment 10 times. We applied a post-hoc clipping, $\max(\hat{\lambda}(t),10^{-4})$, to the estimated intensity function $\hat{\lambda}(\cdot)$. The results, summarized in Table F8 in Appendix F.7 (please see the rebuttal revision manuscript), show that the predictive performance of the point process model indeed improves as the estimation accuracy of the latent triggering kernel increases. We will include this result in the revised manuscript.

---

### Official Review · Reviewer_MBXV · 2025-11-10

**Soundness:** 2
**Presentation:** 2
**Contribution:** 2
**Rating:** 2
**Confidence:** 3

**Summary:**

This paper proposes a new representer-theorem–based method for estimating triggering kernels of linear multivariate Hawkes processes in an RKHS framework. By formulating a system of simultaneous integral equations, the optimal estimator is expressed as a linear combination of transformed kernels evaluated at data points, with all dual coefficients analytically fixed to one. As a result, the method avoids solving high-dimensional optimization problems and achieves substantial computational efficiency gains. Experiments on synthetic data show that the approach matches state-of-the-art predictive accuracy while being significantly faster than existing nonparametric kernel estimators.

**Strengths:**

This paper establishes the first representer theorem for the non-approximated penalized least squares formulation of linear multivariate Hawkes processes, filling a clear theoretical gap in kernel-based point process modeling. The resulting estimator has all dual coefficients analytically fixed to one, eliminating the need for high-dimensional nonlinear optimization and dramatically improving scalability over prior methods. The proposed approach yields a closed-form solution of the integral equations via random feature approximations, avoiding Riemann-sum discretization used in earlier work and enabling efficient computation. The final estimator consists only of additive matrix operations and a matrix inversion whose size is independent of the data size, making the method lightweight, scalable, and well-suited for large-scale Hawkes process datasets. Overall, the paper offers both theoretical novelty and practical computational advantages, advancing kernel-based learning for multivariate Hawkes processes.

**Weaknesses:**

Although the proposed method offers strong computational advantages, it relies on the identity link function and therefore applies only to linear Hawkes processes; models requiring non-linear link functions to enforce non-negativity or capture inhibitory effects fall outside its scope. In addition, the representer theorem is developed under a penalized least-squares formulation rather than maximum likelihood, which may limit statistical efficiency in some settings. The approach also requires solving Fredholm integral equations whose accuracy depends on the random feature approximation, potentially introducing approximation error relative to exact kernel evaluations. Finally, the empirical evaluation is conducted on synthetic datasets, leaving open questions about robustness and performance on real-world event data.

**Questions:**

1. The analytical framework appears to rely on classical techniques, and at first glance the contribution seems incremental. Could you clarify precisely where the conceptual or technical novelty lies in your representer-theorem formulation for Hawkes processes?
2. In the era of deep neural networks, kernel methods are sometimes viewed as outdated or less competitive. Why is a kernel-based approach appropriate and timely for ICLR, and what advantages does it offer over contemporary deep-learning-based models for point processes?
3. The representer theorem is a well-established and classical concept in kernel methods. How does the representer theorem presented in this paper differ from existing versions, including variants used in prior work on point processes? Is the contribution more than a minor extension, and in what sense is it a meaningful and substantial generalization?

---

> ### Author Response · Authors · 2025-11-21
> **Official Comment by Authors [1/2]**
>
> We sincerely thank the reviewer for the valuable comments, to which we provide a detailed response below. Especially, we believe that we can fully address the reviewer's primary concern regarding the paper's novelty by clarifying the difference between our derived representer theorem and conventional ones.
>
> **models requiring non-linear link functions to enforce non-negativity or capture inhibitory effects fall outside its scope**
>
> We believe this concern stems from a misunderstanding. Our proposed method can capture inhibitory effects, as demonstrated in our experiments on the refractory scenario. Please refer to the estimated trigger kernels on the diagonal of Figure A2, which clearly display inhibitory patterns. Our method can enforce non-negativity of the intensity function via post-hoc clipping, such as $\max(\hat{\lambda}(t), \epsilon)$ for a small $\epsilon$, applied to the estimated intensity function $\hat{\lambda}(\cdot)$. This post-processing step has a negligible impact on computational efficiency. We acknowledge that our original explanation may have been unclear, and in the revised manuscript, we will explicitly state that our method is capable of estimating inhibitory effects.
>
> **the empirical evaluation is conducted on synthetic datasets, leaving open questions about robustness and performance on real-world event data.**
>
> This concern was raised by all reviewers. In response, we have added an experiment on a real-world dataset, demonstrating that our approach achieves robust performance while remaining computationally efficient. For further details, please refer to our first response to Reviewer EXZy.
>
> **The analytical framework appears to rely on classical techniques, and at first glance the contribution seems incremental. Could you clarify precisely where the conceptual or technical novelty lies in your representer-theorem formulation for Hawkes processes?**
>
> As discussed in Section 1 (Introduction), the representer theorem for kernel methods is well established in the i.i.d. setting, most notably through the generalized representer theorem by Schölkopf et al. (2001). However, this classical result does not apply when the loss function involves an integral of a latent function, as is the case for point processes. In such settings, establishing rigorous representer theorems remains an important and actively studied open problem in the field.
>
> For non-homogeneous Poisson processes, a representer theorem was obtained only around 2017. In contrast, for Hawkes processes, which constitute a substantially more complex modelling framework because multiple event sequences dynamically interact with each other, no exact (i.e., non-approximate) representer theorem has been established to date, to the best of our knowledge. Our paper is, therefore, the first to rigorously prove an exact representer theorem for linear Hawkes processes. We believe that this provides clear conceptual as well as technical novelty, as it extends the scope of representer theorems to an important and widely used class of mutually interacting point process models.
>
> **The representer theorem is a well-established and classical concept in kernel methods. How does the representer theorem presented in this paper differ from existing versions, including variants used in prior work on point processes? Is the contribution more than a minor extension, and in what sense is it a meaningful and substantial generalization?**
>
> As we already emphasized in our third response above, to the best of our knowledge, this paper provides the first rigorous representer theorem for Hawkes processes. Given the widespread importance of Hawkes processes in applications, we believe it is clear that this contribution is both meaningful and substantially more than a minor extension.
>
> We now clarify two key differences from the previous representer theorem for point processes. First, while Flaxman et al. (2017) established a representer theorem for estimating the intensity function, we establish a representer theorem for estimating the triggering kernel that encodes interactions between events. Although the triggering kernel and the intensity function are closely related, they are distinct objects; our result is therefore qualitatively different from, and not a mere generalization of, the earlier work. Second, in Flaxman et al. (2017), the equivalent kernel appearing in the representer theorem is defined via a single integral equation, whereas the equivalent kernels in our result are defined through a system of integral equations. To the best of our knowledge, this is the first example in the long history of kernel methods where a representer theorem holds under RKHS kernels defined by a system of integral equations. We therefore view our result as unique and potentially impactful for the kernel methods community.

---

> > ### Author Response · Authors · 2025-11-21
> > **Official Comment by Authors [2/2]**
> >
> > **In the era of deep neural networks, kernel methods are sometimes viewed as outdated or less competitive. Why is a kernel-based approach appropriate and timely for ICLR, and what advantages does it offer over contemporary deep-learning-based models for point processes?**
> >
> > We thank the reviewer for raising this fundamental and important question. We agree that deep neural network (DNN) models currently dominate much of the machine learning literature. However, for the following reasons, a kernel method-based approach remains both appropriate and timely for ICLR.
> >
> > * **Reliability of optimization:** Kernel method-based models (including our proposed method) typically reduce to convex optimization problems, which enjoy a unique global optimum and stable training behavior. As a result, they are not subject to the reproducibility issues often observed in contemporary DNNs, where the final performance can vary significantly depending on the optimization algorithm, its tuning parameters, and the parameter initialization. Moreover, kernel method-based models involve only a few regularization parameters and a small number of kernel parameters, resulting in significantly fewer hyperparameters than DNNs. This makes the models substantially easier to tune in practice.
> >
> > * **Model interpretability:** Many DNN-based Hawkes processes (or more generally, history-dependent temporal point processes) embed the past event history into a latent vector via a DNN and directly output the conditional intensity of the next event (e.g., Transformer Hawkes Process (Zuo et al., 2020), Neural Recurrent Marked Temporal Point Process (Duo et al., 2016)). While this approach is powerful especially for marked point processes with unstructured marks and complex history dependence, the influence of past events on future events is effectively a black box, making the event-generation dynamics difficult to interpret or explain. In contrast, both conventional kernel method-based models and our proposed method explicitly learn trigger kernels, which provide a transparent description of how past events affect future intensity functions, thereby offering high interpretability. Interpretability of models is also an important topic for ICLR.
> >
> > * **Complementarity of kernel methods and deep models:** We also view it as unproductive to position kernel methods and DNNs as fundamentally competing paradigms. Hybrid approaches such as deep kernel learning, which combine the computational advantages of kernel methods with the representational power of DNNs, are promising directions for future research. Our current work contributes to this broader agenda by strengthening the theoretical and methodological foundations on the kernel side, which can subsequently be integrated with deep architectures.
> >
> > We will incorporate this discussion into the revised manuscript. We believe that it alleviates the reviewer's concern.

---

> > > ### Comment · Reviewer_MBXV · 2025-11-25
> > >
> > > Thank you very much for your thorough and constructive responses to my comments and questions. Your explanations have addressed my concerns satisfactorily, and I now have a much clearer understanding of the contributions and scope of the proposed framework.
> > > In light of these improvements and clarifications, I have raised my evaluation of the paper.

---

> > > > ### Author Response · Authors · 2025-11-27
> > > >
> > > > We thank the reviewer so much for carefully reading our responses and for raising the evaluation of our work. We understand that the paper is still regarded as marginally below the acceptance threshold. If there remain any unclear aspects of our contributions or of our explanations, we would be very grateful if the reviewer could kindly point them out, so that we may address them as clearly as possible during the rebuttal period.

---

### Official Review · Reviewer_EXZy · 2025-11-10

**Soundness:** 3
**Presentation:** 2
**Contribution:** 3
**Rating:** 6
**Confidence:** 3

**Summary:**

This paper proposes a new representer theorem for linear multivariate Hawkes processes trained with penalized least squares in an RKHS. The theorem says that each triggering kernel can be written as a sum of equivalent kernels** that solve a system of Fredholm integral equations. A key outcome is that the dual coefficients are all 1, so there is no separate optimization over them. The authors show an efficient construction using random Fourier features, which reduces learning to one matrix inversion of size $MU \times MU$ (independent of the number of events). On synthetic datasets, the method keeps similar accuracy to strong baselines while being much faster. The paper also discusses limits: linear link (no guaranteed non-negativity) and cubic cost in the process dimension $U$.

**Strengths:**

- **Originality:** First representer theorem for the non-discretized penalized LS problem in linear Hawkes; fixing all dual coefficients to 1 removes a costly optimization step.
- **Quality:** Clear path from theorem to algorithm using degenerate kernels / random Fourier features, with closed-form estimators.
- **Clarity:** Assumptions and complexity are explicit; the narrative from theory to implementation is easy to follow.
- **Significance:** Very large speed-ups compared to strong baselines while keeping similar accuracy for larger datasets.

**Weaknesses:**

- **Evidence scope:** Only synthetic data are used; real-world robustness and data issues are not tested.
- **Proof placement:** The main-text sketch uses inverse operators and delta distributions and may feel less rigorous; the appendix version is better suited for details.
- **Modeling limit:** Linear link does not guarantee non-negativity of intensity; post-hoc clipping or other fixes may be needed.
- **Scalability in $U$:** Cost scales as $O(N^2 M^2 U^2 + M^3 U^3)$; very high-dimensional processes or very long sequences can still be heavy.

**Questions:**

1. For large \(U\), do **conjugate-gradient** solvers or block preconditioners help? Please provide scaling curves vs. \(U\).
2. How sensitive are results to the **support window $A$**? Can $A$ be learned ?
3. Consider moving the long proof to the **appendix**, and keeping only a short, readable proof sketch in the main text.

---

> ### Author Response · Authors · 2025-11-21
>
> We would like to thank the reviewer for the constructive comments. Below, we provide a detailed response to each of the comments.
>
> **Evidence scope: Only synthetic data are used; real-world robustness and data issues are not tested.**
>
> This concern was raised by all reviewers. In response, we have added an experiment on a financial dataset that is widely used for evaluating Hawkes process models (Du et al., 2016). This dataset contains transaction records of a single stock over one day, with two event types ($U = 2$): “buy” and “sell”. The event sequence is further partitioned by timestamps. From the 100 sequences available on the GitHub repository of (Zuo et al., 2020), each of which size ($N$) is 3319, we randomly constructed 10 pairs of sequences; for a pair, one was used for model training, and the other was used to evaluate the negative log-likelihood as the predictive error (the lower, the better). For our proposed method, we applied a post-hoc clipping, $\max(\hat{\lambda}(t),10^{-2})$ to the estimated intensity function $\hat{\lambda}(\cdot)$. For the models except Exp, the support window $A$ was set at 3. Table F5 in Appendix F.4 (please see the rebuttal revision manuscript) summarizes the results: compared to the baseline methods, our approach achieves robust performance while remaining computationally efficient. We will include this experiment on real-world data in the revised manuscript.
>
> Du et al., Recurrent marked temporal point processes: Embedding event history to vector, KDD2016.
>
> Zuo et al., Transformer Hawkes Process, ICML2020.
>
> **Proof placement: The main-text sketch uses inverse operators and delta distributions and may feel less rigorous; the appendix version is better suited for details.**
>
> We thank the reviewer for the helpful suggestion. We will move the proof based on Mercer's theorem, currently located in the appendix, into the main text and relocate the proof using the inverse operator to the appendix instead. We will also add an explanation that the proof relying on the inverse operator is not fully rigorous.
>
> **How sensitive are results to the support window $A$? Can $A$ be learned ?**
>
> We thank the reviewer for bringing this important point to our attention. To the best of our knowledge, there is no prior work that learns $A$ directly from data. However, $A$ can be regarded as a hyperparameter that controls the shape of the triggering kernel and can therefore be selected from data, for example, via cross-validation. In general, if $A$ is too small, estimation methods cannot capture the true shape of the underlying triggering kernel. Conversely, if $A$ is too large, non-negligible values remain in regions where the true triggering kernel is essentially zero, thereby increasing the estimation error. It is worth noting that choosing $A$ too small is substantially more detrimental than choosing $A$ too large; therefore, it is preferable to set $A$ on the larger side.
>
> To verify this behavior, we evaluated the estimation error of the proposed method for $A \in \\{ 1, 2, 5, 10, 20 \\}$, using data from the mutually-exciting scenario ($T$ = $5000$). Table F6 in Appendix F.5 (please see the rebuttal revision manuscript) summarizes the results, suggesting that an optimal value of $A$ exists. We will include this discussion in the appendix of the revised version.
>
> **Consider moving the long proof to the appendix, and keeping only a short, readable proof sketch in the main text.**
>
> We thank the reviewer for the constructive comments. Following the reviewer's suggestion, we will retain only a concise, readable proof sketch in the main text and relocate the full proof to the appendix.
>
> **Scalability in $U$: Cost scales as $O(N^2 M^2 U^2 + M^3 U^3)$; very high-dimensional processes or very long sequences can still be heavy.** ... **For large (U), do conjugate-gradient solvers or block preconditioners help? Please provide scaling curves vs. (U).**
>
> Please allow us a few more days to prepare our response.

---

> ### Author Response · Authors · 2025-11-24
>
> **Scalability in $U$: Cost scales as $O(N^2 M^2 U^2 + M^3 U^3)$; very high-dimensional processes or very long sequences can still be heavy.** ... **For large (U), do conjugate-gradient solvers or block preconditioners help? Please provide scaling curves vs. (U).**
>
> We would first like to emphasize that the regimes of high dimensionality ($U \gg 1 $) and very long sequences ($N \gg 1$) should be considered separately. For very long sequences ($N \gg 11$), our method exhibits substantially better scalability than the existing nonparametric approach (Bonnet), with a computational complexity of $O(N^2)$ compared to $O(N^4)$. This represents a clear advantage of our approach.
>
> By contrast, in the high-dimensional regime ($U \gg 1 $), our method requires solving a system of linear equations, which incurs a naïve computational cost of $O(U^3)$. In this respect, as the reviewer correctly points out, our approach can indeed become less favorable than Bonnet, whose complexity is $O(U^2)$.
>
> We conducted an additional experiment to empirically characterize how the computational cost of our method scales with respect to $U$. The computational burden of solving the linear system is most evident in settings where the number of events per dimension is small while $U$ itself is large. Accordingly, we set $\mu_i = 0.1$, $g_{ij}(\cdot) = 0$, and $T = 100$, and generated synthetic datasets for $U \in \\{ 10, 50, 100, 200, 300, 500 \\}$. We then measured the training time of the proposed method. Table F7 and Figure F1 in Appendix F.6 (see the rebuttal revision manuscript) summarize the results: empirically, the training time grows sub-cubically in $U$ up to around $U \simeq 100$, and the scaling curve gradually approaches $\mathcal{O}(U^3)$. At least up to $U = 500$, the observed scaling is therefore better than $\mathcal{O}(U^3)$. We attribute this behavior largely to the highly optimized implementation of Cholesky factorization in TensorFlow, which efficiently exploits the 12-core CPU architecture.
>
> Additionally, we implemented a naïve conjugate gradient (CG) solver to compare its runtime with the Cholesky factorization (CF) approach, using a very simple preconditioner given by the inverse of the diagonal of $(\frac{1}{\gamma} I_{MU}+\Xi)$. Unfortunately, this configuration led to worse runtime than CF, suggesting that a substantially more suitable preconditioner is required to fully benefit from CG. Designing such a preconditioner is nontrivial and beyond what can reasonably be accomplished within the rebuttal period. Based on these findings, our current model should be regarded as being practically targeted at event data with up to a few hundred dimensions. We will explicitly acknowledge this point as a limitation of the present work in the revised manuscript and highlight it as an important direction for future research. We hope these adequately address the reviewer's concern.

---

### Meta-Review · Area_Chair_HhDZ · 2026-01-16

**Summary:**

This paper establishes the first exact representer theorem for linear multivariate Hawkes processes under non-discretized penalized least squares. Reviewers generally agreed on the theoretical novelty, clarity of the functional analysis, and clear computational gain on small scale problems.  Concerns about realism (linearity, non-negativity) and empirical scope were mitigated in the rebuttal. Given the clear theoretical contribution, expert endorsement of novelty, and strong efficiency gains, we recommend acceptance.

**Reviewer Concerns:**

Address: real-world data, post-hoc clipping, scalability

Outstanding: beyond linear model, cubic complexity of dimensionality

**Reviewer Scores:**

MBXV might just forgot to update the scores, as the reviewer clearly said the authors address the concerns.

---

### Decision · Program_Chairs · 2026-01-26

Accept (Oral)